# ClpP/ClpX deficiency impairs mitochondrial functions and mTORC1 signaling during spermatogenesis

Chenxi Guo [1,2✉], Yuan Xiao[1], Jingkai Gu [1], Peikun Zhao[1], Zhe Hu[3], Jiahuan Zheng[3], Renwu Hua [1,2], Zhuo Hai[1], Jiaping Su[1], Jian V. Zhang[2,4], William S. B. Yeung [1,3] & Tianren Wang [1✉]

Caseinolytic protease proteolytic subunit (ClpP) and caseinolytic protease X (ClpX) are mitochondrial matrix peptidases that activate mitochondrial unfolded protein response to maintain protein homeostasis in the mitochondria. However, the role of ClpP and ClpX in spermatogenesis remains largely unknown. In this study, we demonstrated the importance of ClpP/ClpX for meiosis and spermatogenesis with two conditional knockout (cKO) mouse models. We found that ClpP/ClpX deficiency reduced mitochondrial functions and quantity in spermatocytes, affected energy supply during meiosis and attenuated zygotene-pachytene transformation of the male germ cells. The dysregulated spermatocytes finally underwent apoptosis resulting in decreased testicular size and vacuolar structures within the semi-niferous tubules. We found mTORC1 pathway was over-activated after deletion of ClpP/ClpX in spermatocytes. Long-term inhibition of the mTORC1 signaling via rapamycin treatment in vivo partially rescue spermatogenesis. The data reveal the critical roles of ClpP and ClpX in regulating meiosis and spermatogenesis.

[1] Shenzhen Key Laboratory of Fertility Regulation, Reproductive Medicine Center, The University of Hong Kong-Shenzhen Hospital, Shenzhen 518053, China. [2] Center for Energy Metabolism and Reproduction, Shenzhen Institute of Advanced Technology, Chinese Academy of Sciences, Shenzhen 518055, China. [3] Department of Obstetrics and Gynaecology, Li Ka Shing Faculty of Medicine, The University of Hong Kong, Hong Kong SAR 999077, China. [4] Shenzhen Key Laboratory of Metabolic Health, Shenzhen 518055, China. ✉email: chancy9199@sina.com; wtrcmu@126.com

Mitochondria are widely known to play a central role in cellular energy production and supply[1]. The chemical energy produced by oxidative phosphorylation in the mitochondria is stored in adenosine triphosphate (ATP)[2]. The energy generated by mitochondria are crucial for various cellular activities, including gene transcription, protein synthesis, cell differentiation and growth, mitosis and meiosis[3–7]. Mitochondria also provide a platform for metabolic pathways, including tricarboxylic acid cycle, oxidation, and lipid synthesis[8]. The functional versatility and adaptability of mitochondria are based on numerous proteins, protein complexes, as well as stress-response pathways[9]. A mature mammalian spermatozoon contains around 80 mitochondria in the mid-piece[10]. In addition to energy production, mitochondria in spermatozoa are implicated in redox equilibrium and calcium regulation essential for capacitation, sperm motility, acrosome reaction and gamete fusion[11]. Available evidences suggest that mitochondria also play diverse roles in germ cell differentiation and spermatogenesis[12,13]. During spermatogenesis, the mitochondria change their morphology, size and location[14]. The spermatogonia contain evenly distributed ovoid shaped mitochondria with lamellar cristae and an electron translucent matrix. The zygotene and early pachytene spermatocytes contain increasing number of elongated mitochondria, signifying an active functional state, in close proximity to the nuclear membrane[14,15]. The late spermatocytes and early spermatids contain mitochondria with enlarged electron-dense matrix and vesicular cristae, evenly distributed throughout the cytoplasm[16]. However, the exact roles of mitochondria in spermatogenesis remain largely unknown. ClpP and ClpX are mitochondria-specific quality control proteases. They maintain proteostasis via degrading misfolded or damaged proteins in mitochondrial matrix[17]. Maintaining protein homeostasis is a challenging work for the mitochondria. Disruption to proteostasis is usually caused by local reactive oxygen species (ROS) produced during normal and attenuated oxidative phosphorylation (Curtis et al.[18]). The ClpP and ClpX complex (ClpXP complex) is a AAA + protease that uses the energy of ATP binding and hydrolysis to degrade unfolded or misfolded proteins[19]. The complex consists of a hexamer of ClpX and a tetradecamer of ClpP[20]. ClpX recognizes the protein substrates via binding with their unstructured peptide sequences, termed as degradation tags or recognition signals[21]. ClpP acts as a proteolytic component of ClpX or other AAA+ proteases; ClpP binds to its partner ATPase and cleaves any polypeptide that is translocated to the proteolytic chamber of ClpP. The small fragments of the cleaved polypeptides can then exit the chamber and be further degraded by exopeptidases to free the amino acids of the fragments[20,22].

Recent studies investigated the role of ClpP and ClpX in germ cell development and reproductive diseases[23–25]. A case study reported a homozygous missense mutation of ClpP in an azoospermic man[26]. ClpP mutations also cause the autosomal recessive Perrault syndrome (PRLTS), which is characterized by primary ovary insufficiency in females and early-onset permanent hearing loss in males and female[27,28]. These observations associate ClpP deficiency with blockage of germ cell differentiation and development, though the underlying mechanism is unknown. The research on ClpX is limited. ClpX can exist as an ATP-dependent molecular chaperone alone, rather than as a ClpXP complex in some species, like yeast, indicating that the ClpXP complex is dispensable in some organisms[29]. The role of ClpX in germ cell differentiation and development awaits further investigation.

Here, we utilized the Cre-LoxP system and generated conditional knock-out (cKO) mouse lines with specific knock-out of ClpP or ClpX in the postnatal germ cells. Examination of the histology and structure of the testis in these mice showed that the deletion attenuated zygotene-pachytene transformation of the spermatocytes. We performed high-throughput sequencing on highly purified spermatocytes and found that the mTOR signaling was significantly altered after genetic deletion of the *Clpp* or *Clpx* gene. Interestingly, in vivo inhibition of mTORC1 can partially rescue the phenotypes. Overall, our data provide important insights into mitochondrial quality control in meiosis and spermatogenesis, and reveal a critical role of ClpP/ClpX in spermatogenesis via regulating the mTORC1 signaling.

## Results

**ClpP or ClpX deficiency decreases testicular size and disrupts spermatogenesis.** Here, we utilized *ClpP^fl/fl* or *ClpX^fl/fl* animals as the WT controls. The *ClpP^fl/fl*;*S8-Cre* or *ClpX^fl/fl*;*S8-Cre* mice were termed ClpP cKO or ClpX cKO mice, respectively hereafter. We sacrificed animals from the same littermate for comparison in majority of experiments. The levels of ClpP and ClpX protein in the purified spermatocytes were determined. We use semi-quantitative western blotting analysis (Fig. 1b) showed that the ClpP cKO spermatocytes had undetectable level of ClpP and significantly higher expression of ClpX relative to the control group (Fig. 1c). Similarly, the spermatocytes of the ClpX cKO mice had almost no ClpX expression, but the expression of ClpP was significantly higher than the control (Fig. 1d). Thus, the data not only demonstrated deletion of the ClpP or ClpX in the respective cKO mice, but also compensation of the ClpP and the ClpX protein after deletion of ClpX and ClpP, respectively. We next checked the testicular size of two cKO mouse lines. The sizes of the testis from both cKO mice were dramatically decrease at PD 56 when compared with the controls (Fig. 1e, f). No difference was detected in body weight between the control and the ClpP/ClpX cKO animals (Fig. 1g), while the testis/body weight ratio showed significant decrease in both cKO mice relative to the control mice (Fig. 1h).

Histological analysis of the testis and epididymis tissue from mice at PD 56 demonstrated strong negative effect on spermatogenesis in seminiferous tubules of the mutant animals (Fig. 2a–c). Figure 2a shows images of overall patterns of testicle structure from the control and the mutant mice at low magnification. Severe germ cell loss was observed in majority of the seminiferous tubules in the ClpP cKO testis (Fig. 2a). The cells of the seminiferous tubules were not well organized with loose patches of germ cells and a lot of empty space in the tubules (Fig. 2b). In contrast to the tubules of the controls (*Clpp^fl/fl* or *Clpx^fl/fl*) with germ cells at all stages of spermatogenesis (Fig. 2b), the tubules of the ClpP cKO contained only spermatogonium (SG) and spermatocytes (SC), and almost without round spermatids (ST) and elongated spermatozoa (SZ). The presence of empty areas (red asterisks) in the central and peripheral regions of the seminiferous tubules indicated germ cell loss after ClpP germ cells specific deletion (Fig. 2b).

The ClpX cKO animals exhibited much more severe phenotypes of the seminiferous tubules than the ClpP cKO animals; almost all the germ cells were lost in every tubule under low magnification (Fig. 2a). At high magnification, there were a lot of vacuolar structures in the tubules (red asterisks; Fig. 2b), which were filled with germ cells in the controls (Fig. 2b). The spermatogonia in the ClpX cKO remained in the peripheral region of the seminiferous tubules (Fig. 2b). No sperm was observed in the epididymis of the ClpP cKO and the ClpX cKO mice, while mature sperms filled up the control epididymis (Fig. 2c).

To quantify the difference in testicular morphology, we determined the diameters of the seminiferous tubules, which were similar between the ClpP control (~251.5 μm) and the ClpX control (~259.7 μm). The diameters of the seminiferous tubules in

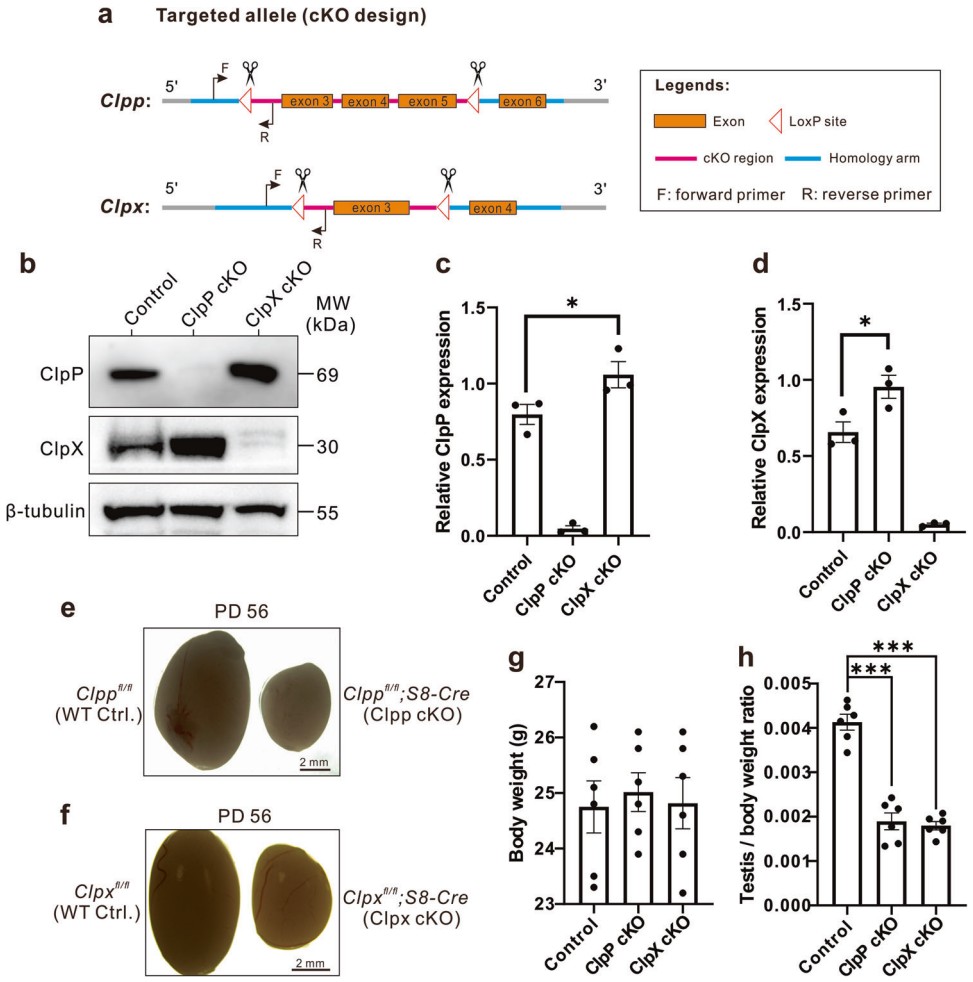

**Fig. 1 ClpP/ClpX conditional knock-out (cKO) mice demonstrate decreased testis size. a** Design of the Cre-LoxP cKO system in the *ClpP/ClpX* allele, such that exon 3–5 was flanked by two LoxP regions in the *ClpP* allele, and exon 3 was flanked by two LoxP regions in the *ClpX* allele. When combined with Cre protein, exon 3–5 is deleted to complete *ClpP* gene knock-out; When combined with Cre protein, exon 3 is deleted to complete *Clpx* gene knock-out. The forward and reverse primers were designed to check the insertion of the LoxP site in the *ClpP/ClpX* allele. **b** Western blot analysis to show the level of ClpP/ClpX expression in the spermatocytes of control and ClpP/ClpX cKO mice. β-tubulin was used as a loading control. The expression of (**c**) ClpP and (**d**) ClpX were semi-quantified in three groups. **e, f** Representative examples of the testes isolated from *ClpP^{fl/fl}/ClpX^{fl/fl}* mice and *ClpP^{fl/fl}/ClpX^{fl/fl}; Stra8-Cre* cKO mice, which were collected on postnatal day (PD) 56. The scale bar is 2 mm. Bar charts to show the mean ± SEM **g** body weight and **h** testis weight/body weight ratio in the control and ClpP/ClpX cKO mice groups (*n* = 6 for each group) at PD 56. *$p < 0.05$, ***$p < 0.001$.

the ClpP cKO (~211 μm) and the ClpX cKO (~128 μm) mice were much reduced. In addition, the number of spermatogonium in the ClpX cKO was significantly lower than the other three groups (Fig. 2d), and that the number of spermatocytes (the primary and secondary spermatocytes) in the cKO mice were dramatically reduced when compared with their respective controls (Fig. 2e).

**Deletion of ClpP/ClpX disrupts development and promotes apoptosis of spermatocytes.** The ClpP/ClpX cKO testicular sections were immune-stained for γH2A.X (green; Fig. 3a, b) to label DNA double-strand breaks. The sections were co-immuno-stained with SYCP3 (red) for all chromosomes of meiotic cells and the nuclei were stained with DAPI (blue). The expression of γH2A.X in majority of the spermatocytes in the controls (*ClpP^{fl/fl}* or *ClpX^{fl/fl}* mice; Fig. 3a, b) was concentrated in the sex body, suggesting that these cells were in the pachytene stage or later stages of meiosis. In the ClpP cKO sections, several clusters of spermatocytes in the central zone of the seminiferous tubule retained the γH2A.X signals not only in the sex body but also in

the autosomes (abnormal SC, Fig. 3a), suggesting a disruption of meiosis in the ClpP deficient germ cells. Similarly, the ClpX cKO sections exhibited a great number of abnormal spermatocytes at PD 35 with γH2A.X signal retention in the autosomes (yellow arrow heads, Fig. 3b). The phenotype in the ClpX cKO sections was more severe than that in the ClpP cKO sections.

We further immune-stained the testis sections for H1t, which is normally expressed after the mid-pachytene stage in spermatocytes. The expression pattern of H1t was different between the controls and the ClpP/ClpX cKO mice. More H1t positive cells were observed in the controls than the cKO groups. Majority of the H1t positive cells in the control group were at the late stages of meiosis, including round spermatids and spermatozoa, whereas the H1t positive cells in the ClpP/ClpX cKO mice were bigger in size with pachytene-like appearance and were abnormally aggregated in the seminiferous tubules (Fig. 3c, d). These results suggested that the deletion of ClpP/ClpX disrupted pachytene entry during meiosis.

Since we noticed less germ cells in the ClpP/ClpX cKO testis, we examined apoptosis of germ cells in the mice by immunos-taining for apoptotic markers, cleaved-caspase 3 and cleaved-

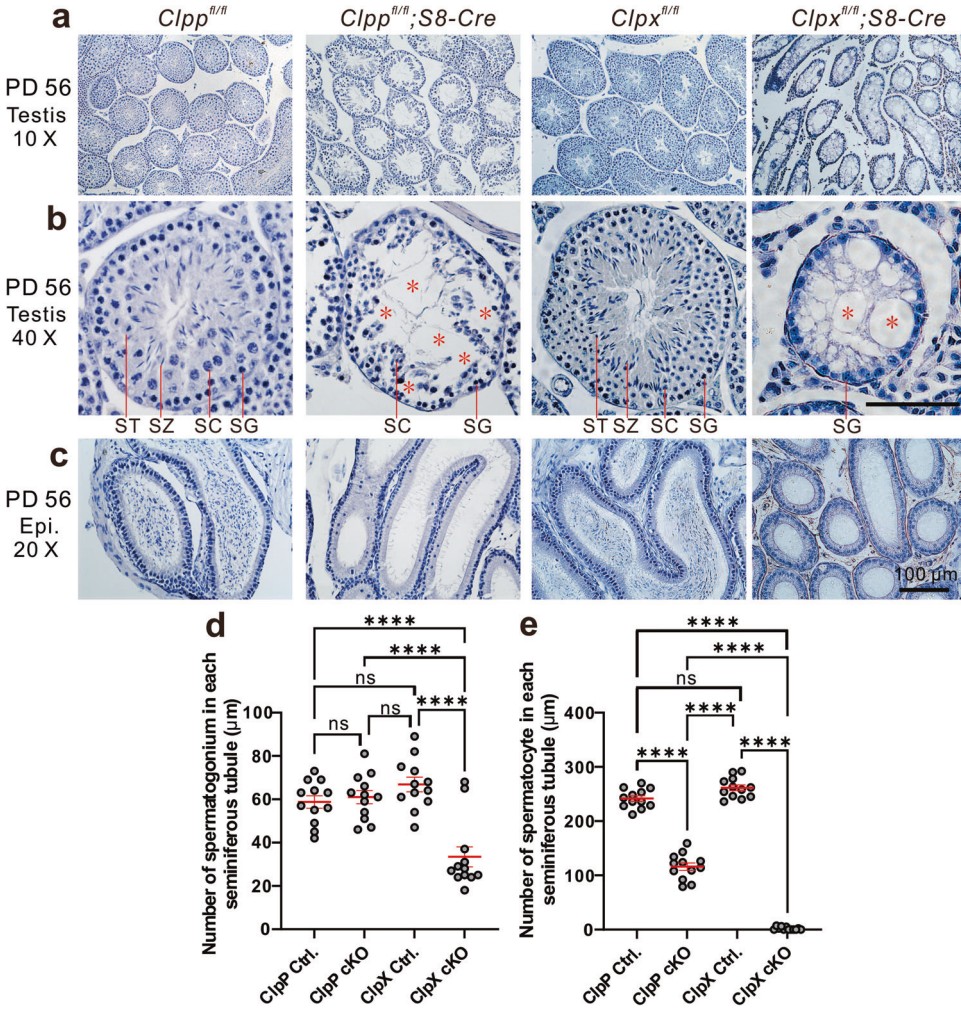

**Fig. 2 The ClpP/ClpX cKO mouse exhibits abnormal histological structure of testis and disrupted spermatogenesis. a, b** Testis sections acquired at PD 56 from *Clpp*^fl/fl, *Clpp*^fl/fl;Stra8-Cre mice and *ClpX*^fl/fl, *ClpX*^fl/fl;*Stra8-Cre* mice, were stained with hematoxylin. In the *Clpp*^fl/fl/*ClpX*^fl/fl;*Stra8-Cre* testes, the red asterisks indicate abnormal tubules in which the majority of the spermatozoon (SZ), spermatocytes (SC) and post-meiotic spermatids (ST) were absent. SG: spermatogonia. **c** Epididymis sections were also stained with hematoxylin reagent to visualize the histological structure and stored mature sperms. Scale bars are 100 μm. **d** Individual dot plot to show the number of spermatogonia in each seminiferous tumble in the ClpP/ClpX control and ClpP/ClpX cKO testes. **e** Individual dot plot to show the number of spermatocytes in each seminiferous tubules in the ClpP/ClpX control and ClpP/ClpX cKO testes. In (**d**, **e**), the data represent the mean ± SEM of three independent experiments. ns means no significant difference. ****p < 0.0001.

PARP (Fig. 3e, f). Compared with the controls, more ClpP/ClpX cKO spermatocytes expressed apoptotic signals (yellow arrow heads; Fig. 3e, f). The number and the percentage of apoptotic germ cells were higher in the ClpP/ClpX cKO sections than their respective control.

**ClpP or ClpX is required for maintaining meiotic process**. We utilized meiotic spread assays to determine whether the key activities during meiosis were disrupted after knocking out of ClpP or ClpX. The spermatocytes were isolated and divided into 4 meiotic stages, namely leptotene, zygotene, pachytene and diplotene (Fig. 4a). In the control group (*Clpp*^fl/fl or *Clpx*^fl/fl), the γH2A.X signals were expressed widely in all chromosomes during the leptotene and zygotene stage. The signal became concentrated only in the sex chromosome when entry into the pachytene stage and this specific expression pattern was retained through the diplotene stage. We noticed that the γH2A.X signal could not be concentrated in the sex body and remained at a pachytene-like stage in both cKO mice. Moreover, the strong retention of the γH2A.X expression was also found in the autosomes of the

diplotene-like stage (Fig. 4a). The percentages of cells at different meiotic stages in the cKO and control mice were compared. Majority of the ClpP/ClpX cKO spermatocytes stuck during transformation from the zygotene to the pachytene stage (Fig. 4b).

We next immune-stained the spermatocytes for MLH1 to check whether crossover was affected. In the control pachytene spermatocytes, the chromosomes were positively stained with the anti-SYCP3 antibody (red), and the MLH1 foci were expressed on each chromosome (green, Fig. 4c), illustrating normal crossover activities during meiosis. However, almost no MLH1 foci was detected on the chromosomes in the pachytene-like stages of the ClpX cKO spermatocytes. The number of MLH1 foci in the ClpP cKO spermatocytes was remarkably decreased (Fig. 4c). We further quantified the number of MLH1 foci in each cell. The data showed significant decrease in the ClpP/ClpX cKO cells relative to the controls, and that the decrease was more severe in the ClpX cKO cells than the ClpP cKO cells (Fig. 4d).

Next, we tested whether the chromosomal synapsis was affected in the ClpP/ClpX deleted spermatocytes. The meiotic spread assays with immune-staining for a synapsis specific

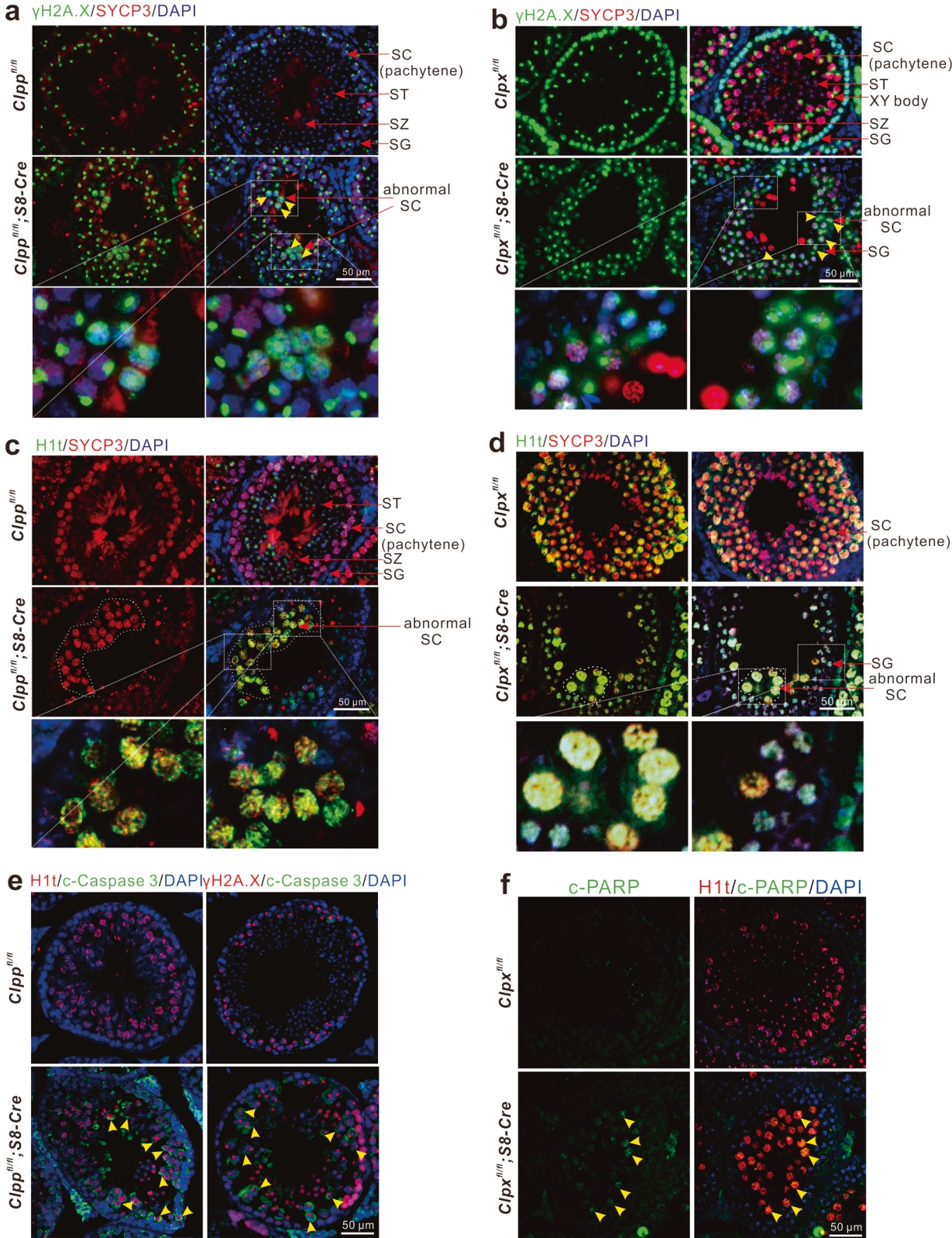

**Fig. 3 ClpP/ClpX cKO mouse exhibits abnormal spermatocyte differentiation. a–f** Testis tissue sections from controls and cKO groups were immunolabelled with various primary antibodies (γH2A.X labels double-strand breaks, SYCP3 labels meiotic spermatocytes, H1t labels mid-pachytene spermatocytes, cleaved-Caspase3 or cleaved-PARP labels apoptotic cells and DAPI labels nuclei) and they were co-stained with DAPI (blue). All images were acquired via fluorescent microscope. In **a**, **b**, the yellow arrowheads indicate abnormal spermatocytes. In **e**, **f**, the yellow arrowheads indicate apoptotic cells. Scale bars are 50 μm.

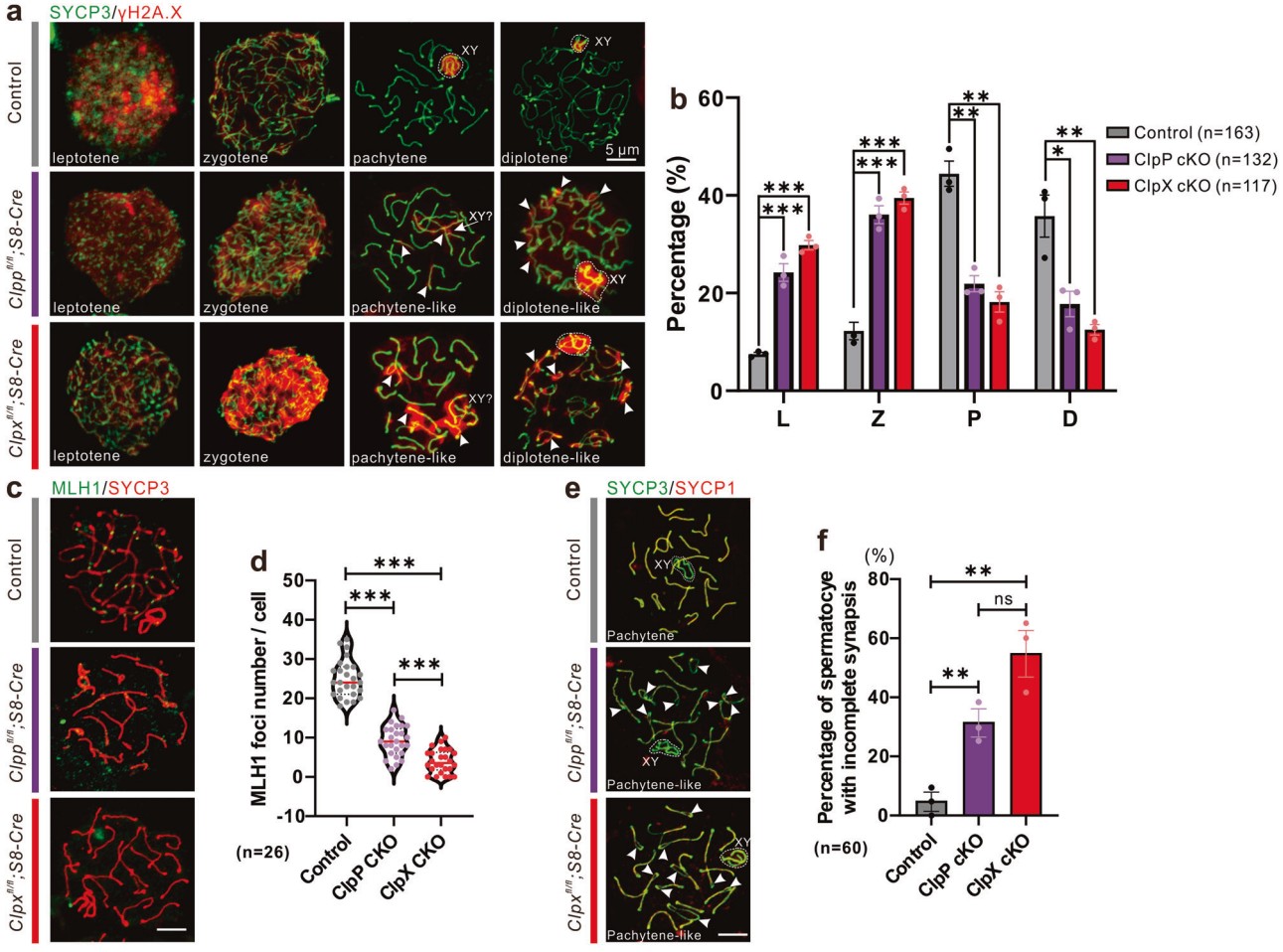

**Fig. 4 Comparison of meiotic chromosome spreads from controls, ClpP cKO and ClpX cKO spermatocytes. a**, **c**, **e** Harsh chromosome spreads were prepared before immunolabeling procedures were conducted with the γH2A.X, MLH1, SYCP1 and SYCP3 antibodies. The SYCP3 antibody was used to label meiotic chromosomes. **a** Spermatocytes are shown at different stages. **b** Bar chart shows the mean ± SEM of percentage of spermatocytes at these stages (i.e., L: leptotene, Z: zygotene, P: pachytene, and D: diplotene) were quantified. $n = 163$ for control spermatocytes, $n = 132$ for ClpP cKO spermatocytes and $n = 117$ for ClpX cKO spermatocytes. **c** Spermatocytes in different groups were immunolabelled with the anti-MLH1 (in green) and anti-SYCP3 (in red) antibodies. **d** The number of MLH1 foci were quantified in the controls, ClpP cKO and ClpX cKO spermatocytes ($n = 26$ for each group). **e** Spermatocytes in different groups were immunolabelled with the anti-SYCP1 (in red) and anti-SYCP3 (in green) antibodies. The dashed line cycled XY body and white arrow heads indicate chromosomes asynapsis. (**f**) Bar chart shows the mean ± SEM of percentage of spermatocytes with incomplete synapsis ($n = 60$ for each group). Scale bars are 5 μm. *$p < 0.05$; **$p < 0.01$; ***$p < 0.001$; ns non-significant difference.

marker, SYCP1 were performed (Fig. 4e). Expression of SYCP1 on each autosome rather than the sex chromosome was observed in the pachytene spermatocytes of the control mice, indicating the occurrence of normal chromosome synapsis. However, the ClpP/ClpX cKO spermatocytes exhibited asynapsis of the autosomes at the pachytene-like stage; the sister chromosomes were separated (white arrow heads) and the expression of SYCP1 (red) was absent (Fig. 4e). Quantification of the percentages of incomplete synapsis at the pachytene/pachytene-like stage showed that 5% of the control pachytene spermatocytes presented asynaptic chromosomes, whereas the percentage increased to 31.6% and 55% in the ClpP cKO and the ClpX cKO spermatocytes, respectively (Fig. 4f). Our data demonstrated the ClpP and ClpX were required for spermatocyte chromosomal synapsis at the pachytene stage, as well as the following crossover activities. The lack of ClpP or ClpX expression in the spermatocytes attenuated the meiotic process, specifically during the zygotene-to-pachytene transition.

**ClpP/ClpX is crucial for telomere-nuclear envelope attachment and α-tubulin formation.** To further understand the role of ClpP/ClpX in meiosis, we investigated the dynamic of telomeres

during meiosis. Mild meiotic spread assay was performed with immunostaining for telomeres and nuclear envelope using the anti-TRF1 antibody and the anti-Lamin B1 antibody, respectively (Fig. 5a). Images of a mid-layer of the control spermatocytes demonstrated nice attachment of the telomeres (TRF1 foci in green) to the nuclear envelope (Lamin B1 labeled membrane in blue), reflecting a normal telomeres-nuclear envelope attachment during meiosis (Fig. 5a). On the other hand, the ClpP/ClpX cKO cells showed remarkable number of telomeres that failed to attach to the nuclear envelope and located in the central region of the nucleus (white asterisks; Fig. 5a). We counted the number of detached TRF1 foci and found significant increase in the number in the ClpP/ClpX cKO cells when compared with the controls (Fig. 5c). The ratio of cells with failure of telomere-nuclear envelope attachment to the total cells was also significantly increased in the ClpP/ClpX cKO spermatocytes (Fig. 5b).

During the experiments, we found that the ClpP/ClpX cKO spermatocytes were fragile when compared with the control spermatocytes. It is known that chromosome and telomere dynamics are associated with cytoskeleton functions[30,31]. Therefore, the cytoskeleton of the ClpP/ClpX cKO cells was checked. The

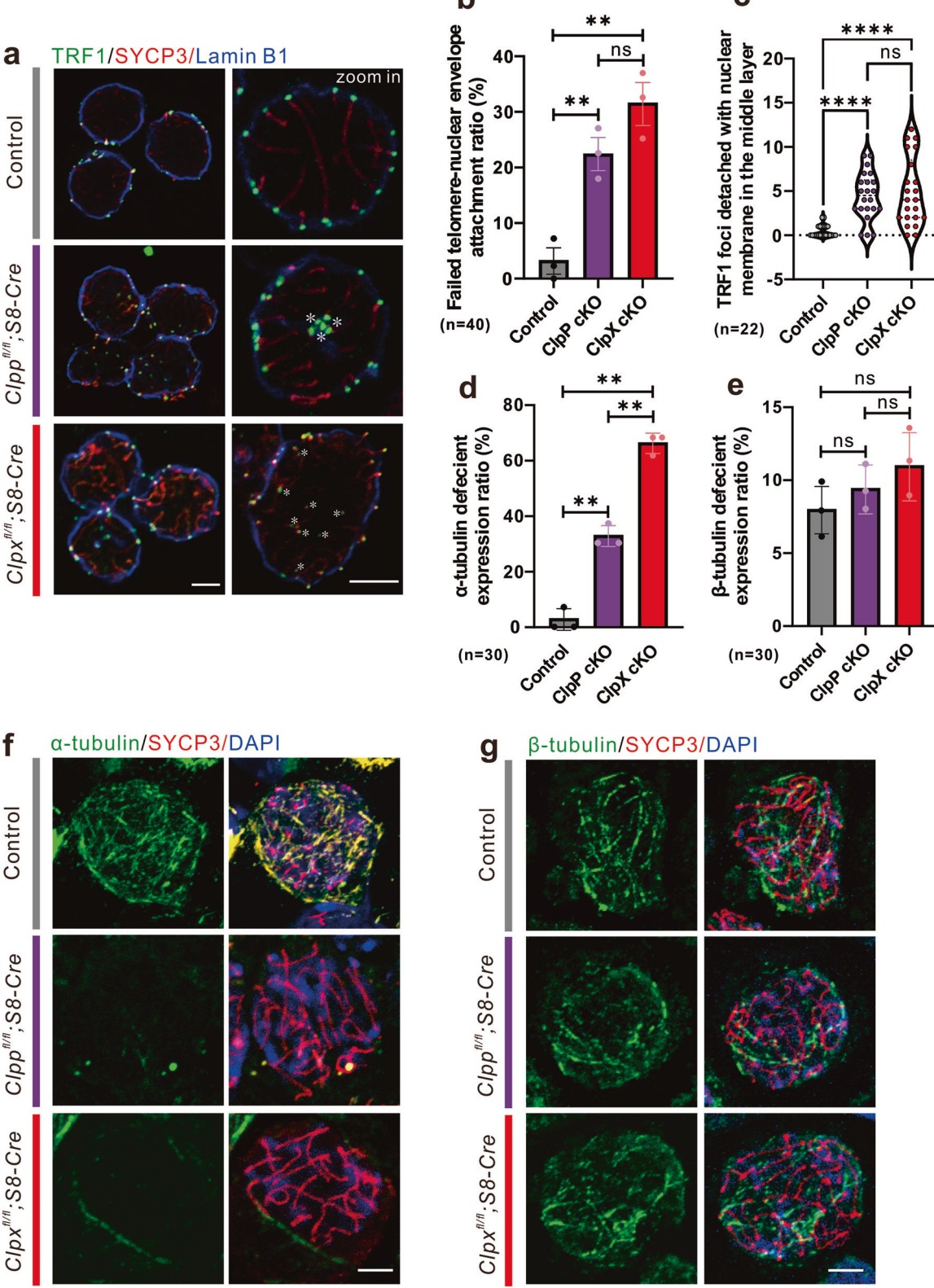

spermatocytes were purified and cytospun onto a glass slide. Immunostaining for α-tubulin and β-tubulin showed that there was no difference in β-tubulin expression intensity when compared with the controls, but the α-tubulin formation was severely attenuated in the ClpP/ClpX cKO spermatocytes (Fig. 5f–g). Attenuation of α-tubulin formation was found in more than 60% of the ClpX cKO spermatocytes and around 35% of the ClpP cKO

spermatocytes; both percentages were significantly higher than that in the control spermatocytes (Fig. 5d). No significant difference was found for β-tubulin formation among these groups (Fig. 5e).

**Deletion of ClpP/ClpX affects mitochondrial quantity and morphology.** We performed electron microscopy (EM) to study the mitochondria in the germ cells. In the control cells, the

**Fig. 5 Evaluation of the attachment between telomere and NE during meiosis and tubulin formation in ClpP/ClpX cKO spermatocytes. a** The spermatocytes from control, ClpP cKO and ClpX cKO groups were prepared to perform mild chromosome spreads before immunolabelling with an anti-TRF1 (in green), anti-SYCP3 (in red) and anti-Lamin B1 (in blue) antibodies, Lamin B1 acts as a marker to label nuclear membrane. White asterisks indicate detached telomeres with NE. Quantification of **b** failed telomere-NE attachment ratio in spermatocytes in each group (n = 40, bar chart shows the mean ± SEM) and **c** the number of TRF1 foci which detached with NE in the middle layer (n = 22 spermatocytes for each group). The spermatocytes with cytospin were immunolabeled with **f** anti-alpha-tubulin and **g** anti-beta-tubulin antibodies to visualize the formation of tubulin-based cytoskeleton in control, ClpP cKO and ClpX cKO spermatocytes. These samples were co-labeled with SYCP3 to confirm the mitotic germ cells. Bar charts show the mean ± SEM of **d** alpha-tubulin or **e** beta-tubulin deficient spermatocyte ratio (n = 30 for each group). All images were captured by confocal microscopy. Scale bars are 5 μm. **p < 0.01; ****p < 0.0001; ns non-significant difference.

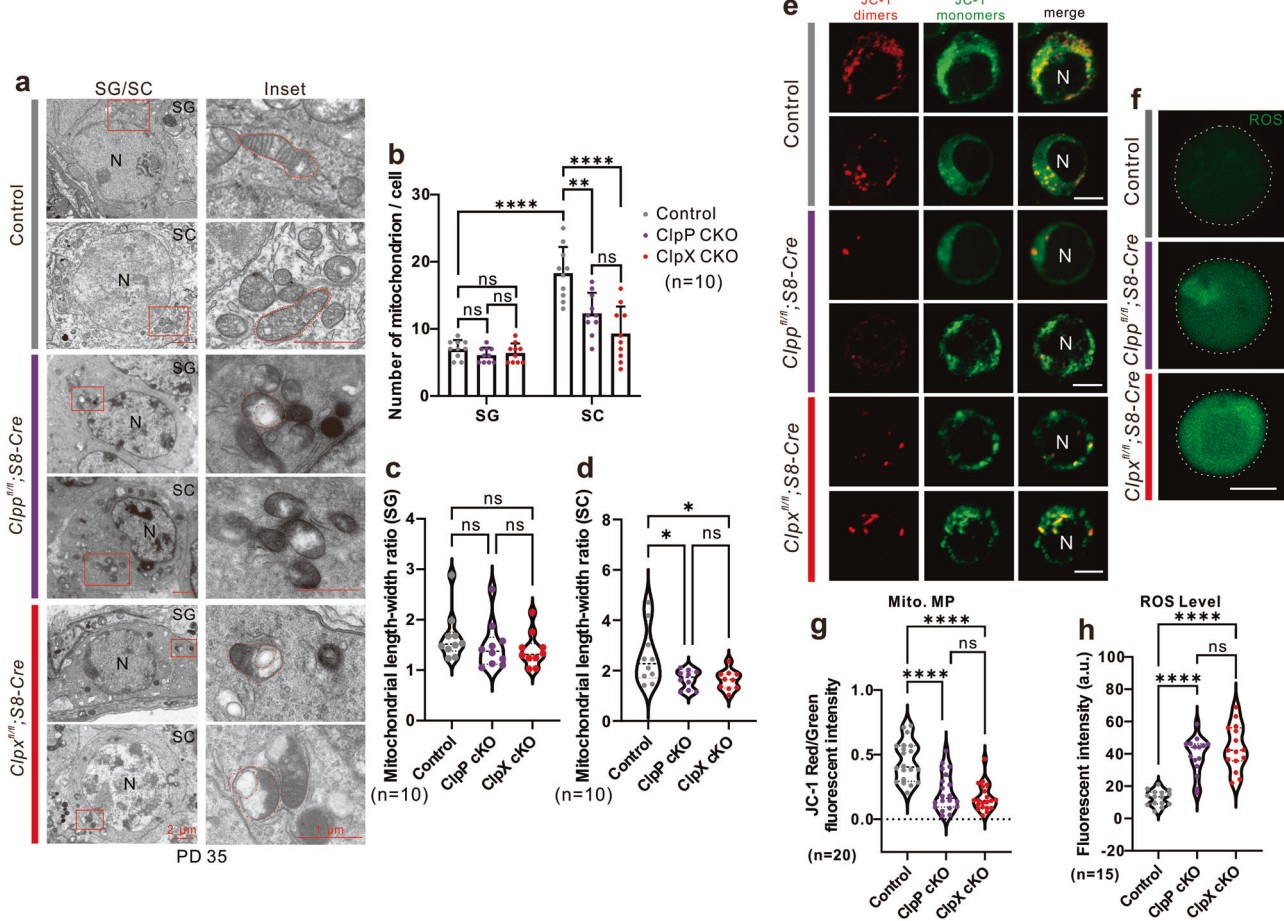

**Fig. 6 Evaluation of mitochondrial expression and functions in germ cells. a** The testis tissue sections were prepared for transmission electro-microscopy (TEM). Spermatogonium (SG) and spermatocytes (SC) were captured to visualize the morphology of mitochondria. N means nucleus. Scale bars are 2 μm in the SG/SC panel and 1 μm in the inset panel, respectively, as labeled in the figure. **b** Number of mitochondria in each cell captured layer were quantified in both SG and SC cells. Mitochondrial length/width ratio was also quantified in **c** SG and **d** SC, respectively. **e** The mitochondrial membrane potential was measured via JC-1 staining in the control, ClpP cKO and ClpX cKO spermatocytes, JC-1 aggregates were labelled in red and JC-1 monomers were labelled in green, N means nucleus, scale bars are 10 μm. **f** The level of ROS was visualized in these three groups via labeling with carboxy-$H_2$DCFDA (in green), scale bar is 10 μm. Images in e-f were captured via confocal microscope. **g** Mitochondrial membrane potential was quantified via the ratio of red/green fluorescent intensity, 20 spermatocytes were quantified in each group. **h** The ROS level of spermatocytes in three groups was quantified via fluorescent intensity, n = 15 for each group. *p < 0.05, **p < 0.01; ****p < 0.0001; ns non-significant difference.

mitochondrial morphology in the spermatogonia (SG, Fig. 6a) and spermatocytes (SC, Fig. 6a) was normal with elongated shape and well-formed inner/outer membranes, intermembrane space and cristae. The mitochondrial morphology in the corresponding ClpP/ClpX cKO cells were different. Almost no elongated mitochondria could be detected in the ClpP/ClpX cKO spermatocytes. In addition, the cristae structure was disrupted and vacuolar structures were observed in the mitochondria (Fig. 6a). The quantification data shows no statistical difference in the number of mitochondria and the mitochondrial length-width ratio

between the controls and the ClpP/ClpX cKO spermatogonia (Fig. 6b, c). However, these parameters were significantly lower in the ClpP and ClpX deficient spermatocytes than the controls (Fig. 6b, d). The WT spermatocytes harbored more mitochondria than the spermatogonia (Fig. 6b).

**Deletion of ClpP or ClpX affects mitochondrial function in spermatocytes.** Next, we studied the effects of ClpP or ClpX deficiency on mitochondrial functions. First, JC-1 staining in

isolated live spermatocytes was conducted. Red fluorescence reflects JC-1 aggregates and green fluorescence reflects JC-1 monomers, and the ratio of red/green fluorescence is a measure of the mitochondrial membrane potential. The control spermatocytes exhibited stronger red fluorescent signals than the ClpP/ClpX cKO spermatocytes, whereas the green fluorescent signals were comparable among them (Fig. 6e). The red/green fluorescence ratios were significantly lower in the ClpP/ClpX cKO spermatocytes than the controls (Fig. 6g), suggesting that the mitochondrial membrane potential was attenuated in the ClpP/ClpX deficient spermatocytes.

We also evaluated the ability of ROS level recovery (Fig. 6f) and found that the ClpP/ClpX deficient spermatocytes could not reduce the ROS level back to the normal value after $H_2O_2$ pretreatment; both groups showed a significant higher ROS level relative to the controls (Fig. 6h). In addition, quantitative PCR assays showed a significant altered expression of several respiratory chain complex genes including Atp5a1, *Ndufv1*, *Cox1*, *Uqcrc2* and *Sdhb* in the ClpP/ClpX cKO spermatocytes (Fig. S1a, b). Based on these observations, we concluded that the deficiency of ClpP or ClpX attenuated the mitochondrial functions in spermatocytes.

**Transcriptome profiling and m6A-seq analysis of ClpP/ClpX cKO spermatocytes**. The germ cells from PD 35 mice were isolated and purified by gravity sedimentation. We selected the pachytene and the pachytene-like spermatocytes (diameter 18–25 μm). RNA sequencing and m6A sequencing were performed on the spermatocytes from the controls, ClpP cKO and ClpX cKO mice (three mice for each group). Principal components analysis (PCA) and other Quality control analysis showed clustering of the 3 groups of samples except the sample WT Ctrl. 3, which was away from the other two control samples WT Ctrl. 1 and 2 (Fig. 7a Fig. S2c–e). When compared with the controls, the ClpP cKO spermatocytes had 1047 down-regulated genes and 147 up-regulated genes, while the ClpX cKO spermatocytes showed 3175 down-regulated genes and 1496 up-regulated genes (Figs. S2a, b and S2f). Gene ontology (GO) enrichment analysis confirmed our observed phenotypes and highlighted changes in spermatogenesis, spermatid differentiation and development in the ClpP cKO and ClpX cKO samples (Fig. 7b, c and Fig. S2h–i). The analysis further suggested that the ion channel activity, DNA packaging complex and nucleosome were affected (Fig. 7b, c). Heatmap of the transcriptome profiles of the 9 samples showed clustering of the WT controls, while the profile of the ClpX cKO 1 was mixed with that the ClpP cKO profiles (Fig. 7d). Among the differential expressed genes between the ClpP cKO and the WT and between the ClpX and the WT, 1005 of them were common in the two comparisons (Fig. 7e). GO analysis showed that these genes were enriched for the process of sperm formation and motility, ion transport, aminoacyl-tRNA biosynthesis, and ADP metabolism. (Fig. 7f).

We also performed m6A-seq of the ClpP/ClpX cKO spermatocytes to evaluate the change in RNA methylation level (Fig. S2j, k). Based on Kyoto Encyclopedia of Genes and Genomes (KEGG) analysis, we found majority of the highlighted genes were enriched in metabolic pathways and both the ClpP cKO and ClpX cKO spermatocytes showed alteration of RNA methylation in mTOR signaling related genes (Fig. 7g, h). The conclusion was supported by the expression levels of the mTOR signaling genes in the controls, ClpP cKO and ClpX cKO groups (Fig. S2g). Based on the heatmap results, majority of the mTOR signaling genes, including *Akt1*, *Rps6*, *Mapk1*, *Rhoa*, etc., were upregulated, whereas around one-third of the mTOR signaling genes, including *Braf*, *Atg13*, *Sspo* etc., were downregulated after deletion of ClpP/ClpX (Fig. S2g).

**mTORC1 signaling is over-activated in ClpP/ClpX deleted spermatocytes**. We confirmed over-activation of the mTORC1 signaling pathway in the ClpP/ClpX cKO spermatocytes by western blotting (Fig. S3). Next, we conducted in vivo pharmacological treatment with a widely known mTORC1 inhibitor, rapamycin. The ClpP/ClpX cKO mice were intraperitoneally injected with PBS (control) or rapamycin (8 mg/kg) every other day from PD 14 and the mice were sacrificed on PD 35 (Fig. 8a). Histological analysis of testis and epididymis tissue failed to observe the round spermatids or spermatozoon, respectively in the ClpP/ClpX cKO mice receiving PBS. Instead, vacuolar structures (red asterisks) probably formed after germ cell loss were observed (Fig. 8b). Surprisingly we found both round spermatids and elongated spermatozoa in the testicular sections of the rapamycin injected ClpP/ClpX cKO mice at PD 35 (Fig. 8b, Fig. S4a, e). The observation indicated that the inhibition of mTORC1 signaling partially rescue the meiotic process of germ cells in the ClpP/ClpX cKO mice, though mature sperms could still not be found in their epididymis (Fig. 8b). Quantification of the histological examination shows a significant increase in the percentage of seminiferous tubules with elongated spermatozoa in the rapamycin treated ClpP/ClpX cKO mice (Fig. 8c). There was no further improvement in the rescue outcome when the rapamycin treatment was extended to PD 56; no mature sperm could be detected in either the testis or the epididymis (Fig. S4d). The sizes of the rapamycin treated cKO testes remained comparable with the PBS injected ClpP/ClpX cKO testes at PD 56 (Fig. S4c). In addition, we found that the seminiferous tubules of the rapamycin-treated tissues were more closely packed than that of the controls (Fig. S4c, top panel).

To confirm the suppression of mTORC1 after rapamycin treatment, we performed western blot analysis on the purified spermatocytes (Fig. 8d). The phospho-mTOR level and pan-mTOR were up-regulated in the ClpP cKO and the ClpX cKO spermatocytes (Fig. 8e), and they returned to level found in the WT cells after rapamycin treatment (Fig. 8d, e). The two main substrates of mTORC1, p70-S6K and S6, were also evaluated. The expression of phospho-p70 S6K was almost undetectable in the ClpP/ClpX WT controls. On the other hand, we detected a remarkable increase in phospho-p70 S6K expression in the ClpP cKO injected with PBS, consistent with the activation of p70 S6K in a ClpP deficient condition (Fig. 8d). The activation was inhibited after rapamycin treatment and the band was much weaker compared with the ClpP cKO group (Fig. 8d). A very weak band was seen in the ClpX cKO+vehicle and the ClpX cKO +rapamycin groups. The pan-p70 S6K levels were comparable among these groups, suggesting that the phosphorylation but not the total amount of p70 S6K was increased in the absence of ClpP or ClpX (Fig. 8d). The pan-S6 level was slightly decreased in the ClpP/ClpX cKO spermatocytes when compared with the WT controls, while the phospho-S6 in ClpP/ClpX cKO groups showed dramatic increase when compared with ClpP/ClpX WT controls. In the rapamycin treated ClpP/ClpX cKO groups, the phospho-S6 levels were significantly inhibited (Fig. 8d, f). Another substrate of mTORC1, 4EBP, was also studied. Both the phospho-4EBP1 and pan-4EBP1 showed dramatic increase in the ClpP/ClpX cKO cells relative to the ClpP/ClpX WT controls, and rapamycin treatment could not decrease the expression of phospho-4EBP1 in the ClpP/ClpX cKO cells, consistent with a previous study[32]. The expression of β-actin showed was comparable in all the groups (Fig. 8d).

Since the expression of phospho-S6K is relatively weak and cannot be accurately compared by western blot analysis, we evaluated the expression pattern of phospho-S6K/S6 in testicular sections by immunostaining. In WT controls, almost no obvious phospho-S6K could be detected in the spermatocytes (Fig. 8g), while the ClpP/ClpX cKO spermatocytes showed strong

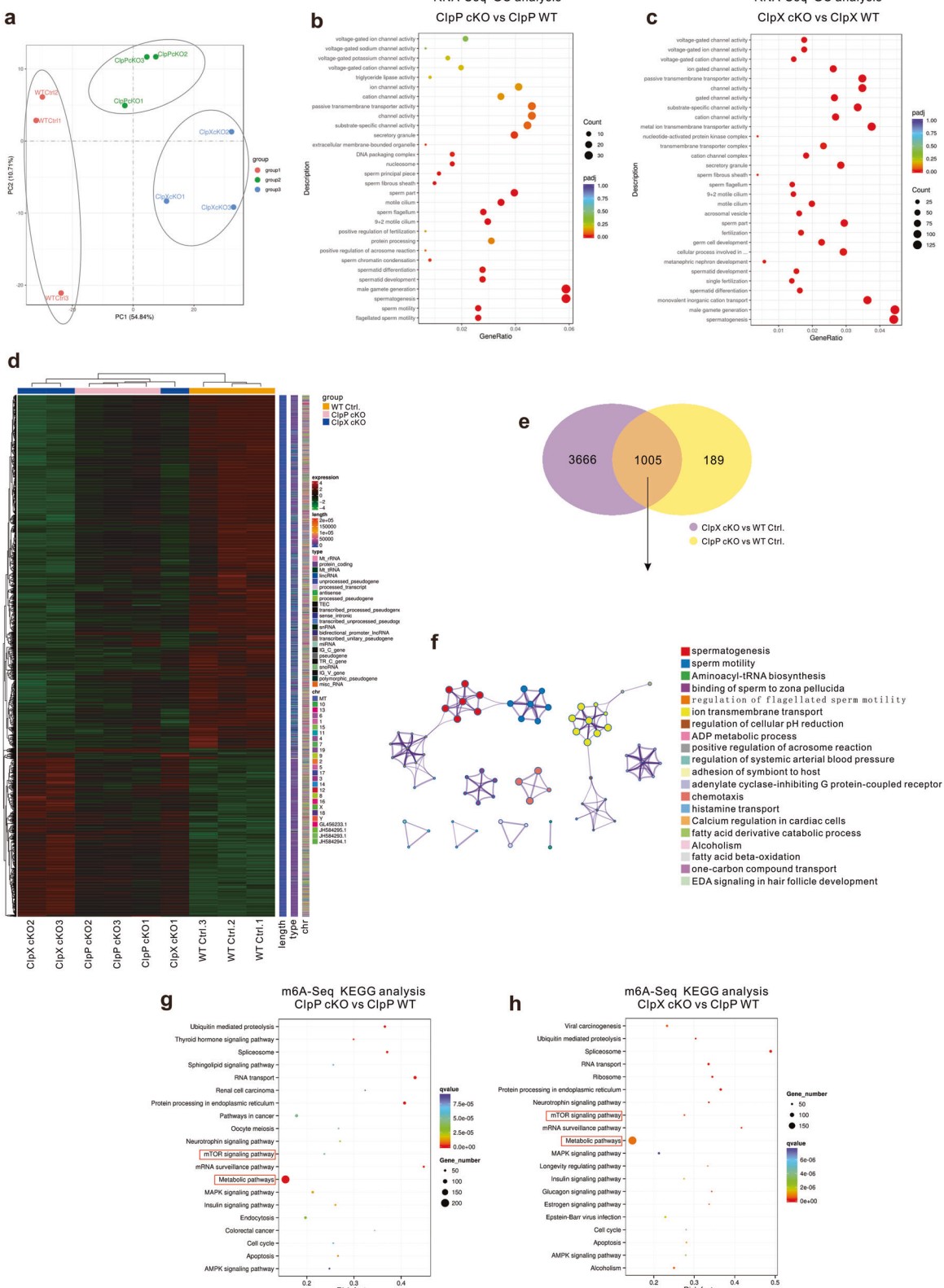

**Fig. 7 Transcriptome profiling and m6A sequencing analysis of spermatocytes in WT control, ClpP cKO and ClpX cKO groups. a** PCA plot of three groups of spermatocytes were shown after RNA sequencing from $n = 3$ mice in each group. The dot plots of GO analysis were shown for **b** ClpP Cko and **c** ClpX cKO when compared with WT controls. **d** Heatmap and cluster analysis of the overview of gene expression patterns in all 9 samples from WT control, ClpP cKO and ClpX cKO groups. **e** Venn analysis shows common changed genes after deletion of ClpP and ClpX in spermatocytes. **f** Metascape network plots showing the relationship of the enriched GO terms in 1005 common genes from Venn analysis. **g**, **h** Spermatocytes were purified from PD 35 WT control or ClpP/ClpX cKO mice testis and sent for m6A sequencing. Dot plots of KEGG analysis showed gene enrichment in related pathways after knocking out of ClpP or ClpX.

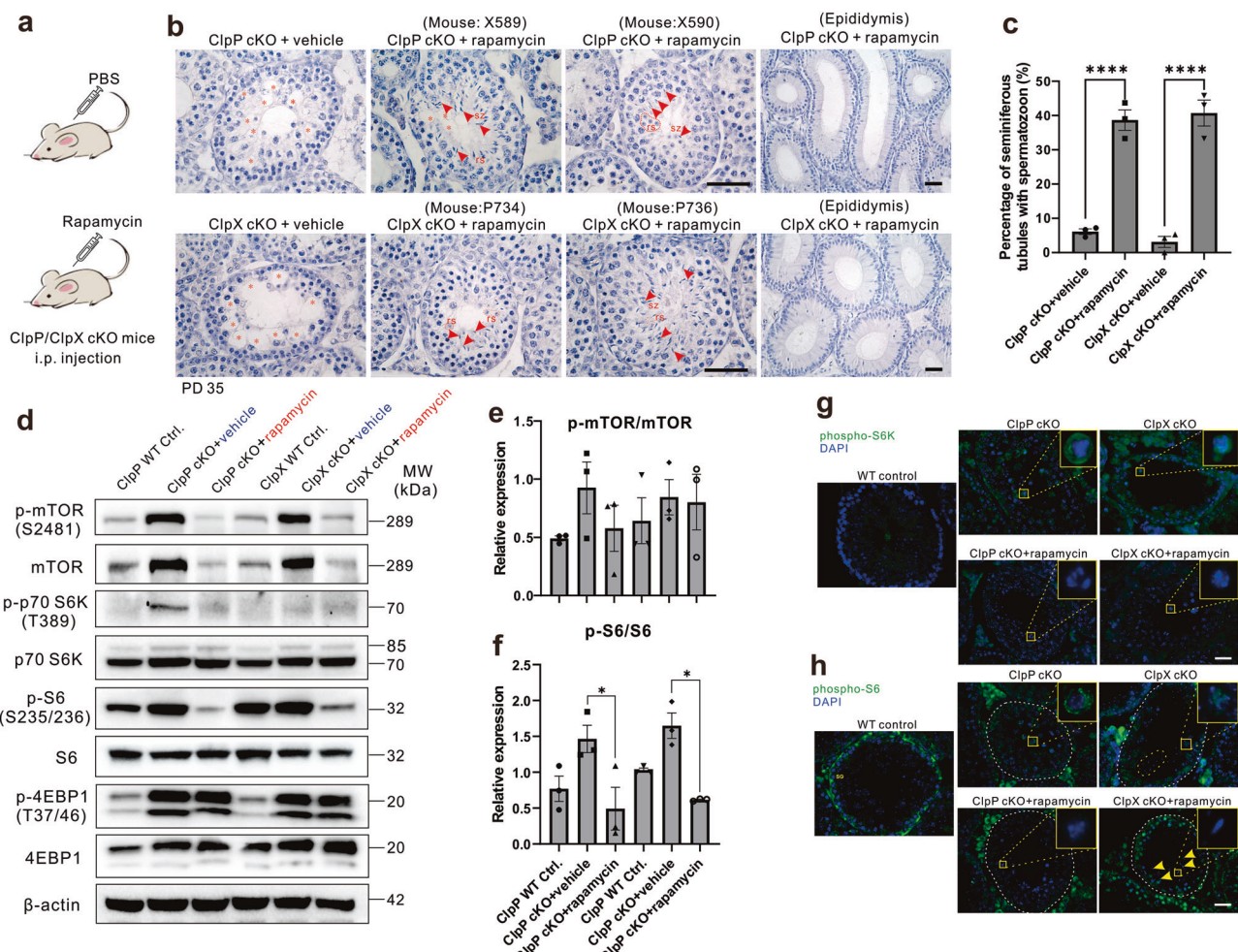

**Fig. 8 In vivo pharmacological treatment of rapamycin to inhibit mTORC1 in ClpP/ClpX cKO mice. a** Carton diagram to describe the plan of drug injection. **b** Histological studies showed PD 35 testis and epididymis sections with hematoxylin staining. X589 and X590 are mice ear tags for ClpP cKO injected with rapamycin, P734 and P736 are mice ear tags for ClpX cKO injected with rapamycin. Red asterisks labeled empty regions with germ cell died out, red arrow heads indicated elongated spermatozoon (sz) and red dashed line circled round spermatids (rs). **c** The percentage of spermatozoon harbored seminiferous tubules was quantified. **d** The protein expression level of mTOR signaling pathways was shown via western blot analysis, the phosphorylation level of **e** mTOR and **f** S6 were quantified. Immunolabeling of **g** phosphor-S6K (in green) and **h** phosphor-S6 (in green) in ClpP/ClpX cKO or ClpP/ClpX cKO injected with rapamycin testis tissue sections, DAPI (in blue) was used to label nucleus. Images were captured via fluorescent microscope. Scale bars are 50 μm. *$p < 0.05$; ****$p < 0.0001$.

cytoplasmic expression of phospho-S6K (green; Fig. 8g). The expression of phospho-S6K in the ClpP/ClpX cKO spermatocytes disappeared after rapamycin treatment (Fig. 8g). Interestingly, we could also see strong phospho-S6 signals in the spermatogonia but not the spermatocytes of ClpP/ClpX WT controls (Fig. 8h), whereas the expression of phospho-S6 decreased in the spermatogonia but increased in the spermatocytes of the ClpP and ClpX cKO testis (Fig. 8h). We also found some fragmented nuclei in the ClpX cKO seminiferous tubule (circled in yellow dotted line, Fig. 8h), suggested apoptosis of the germ cells in the ClpX deficient testes. The phenotype could be partially rescued after rapamycin treatment, with the expression of phospho-S6 in the spermatocytes decreased to levels comparable to the controls and partial recovery of the phospho-S6 expression in the spermatogonia after the treatment. Some elongated spermatozoa were seen (yellow arrow heads, Fig. 8h).

## Discussion
The mitochondrial function is widely recognized as one of the key factors for germ cell development[33–35]. However, the role of

ClpP/ClpX in meiosis and spermatogenesis is not fully understood. Although some recent studies reported ClpP-mediated mitochondrial quality control is important in terms of the ovary aging process[23,24], the role of ClpP/ClpX-mediated mitochondrial quality control in spermatogenesis is still remains unknown. In this study, we utilized Cre-LoxP and *Stra8*-driven Cre system[36] to generate male germ cell specific knockout of ClpP or ClpX in mice. The deficiency of ClpP or ClpX severely affected mitochondrial functions in spermatocytes, which further disrupted meiotic events and progression, and eventually failure of spermatogenesis.

We found genetic compensation of the two genes in the ClpP and the ClpX knockout animals. Compared with controls, the ClpP cKO spermatocytes expressed more ClpX protein, and vice versa for the ClpX cKO spermatocytes. The observation is in line with a previous study showing remarkable ClpX accumulation in many organs, including testis and ovary of ClpP deleted mice[26,37]. The ClpP/ClpX cKO testes were smaller than the controls. Testicular histology revealed that the decreased testis size was due to the loss of spermatocytes. Deletion of ClpP and ClpX affected mainly the spermatocytes and did not have much effect on the

spermatogonia. This differs from our previous study that the deletion of mitofusin 2 protein not only leads to apoptosis of the spermatocytes but also affects survival of the spermatogonia[38].

In this study, we focused on the meiosis process as the loss of spermatocytes could be related to disruption of meiosis. Based on the expression pattern of γH2A.X (DNA double-strand break marker) and H1t (mid-pachytene spermatocyte marker), we found that the ClpP or ClpX deficient spermatocytes could not proceed to the pachytene stage successfully. The observation is in line with a previous study showing failure of progression of spermatogenesis especially at the zygotene-pachytene transformation stage in germ cells of *Mfn1 and Mfn2* mutant mice[39]. Here, our meiotic spread assay showed that majority of the control cells were at the pachytene or diplotene stages while most of the ClpP/ClpX cKO cells were stuck at the leptotene or the zygotene stages on PD 35. Interestingly, we noticed that there are diplotene-like meiotic cells, shared a limited proportion, in *Clpx/Clpp*-deficient spermatocytes. The sister chromosomes in these diplotene-like cells are gradually separated in both ends of chromosomes, while the strong γH2A.X retention in autosomes illustrate these cells are diplotene-like rather than normal diplotene cells. We also interested why we can detect diplotene-like cells in ClpX/ClpP cKO spermatocytes, the reason of it might not be alone. We suggest that although the function of mitochondria got disrupted in ClpX/ClpP cKO spermatocytes, the cell themselves must have their own mechanism to compensate the mitochondrial functions or energy supply to favor the continuation of meiotic process. This could be one possible reason that why we can still detect a little proportion of cells in diplotene-like stage from ClpX/P cKO spermatocytes. Therefore, it also suggests that *Clpx* and *Clpp* might not directly regulate the meiotic process, instead, the deletion of these genes directly damage mitochondria, and affect meiosis in a indirectly way. Moreover, MLH1, which is started to be expressed on chromosomes after crossover activity, it is also a marker protein of the crossover occurring. The lack of expression of MLH1 in both ClpX cKO and ClpP cKO spermatocytes means there is little crossover occurred in the mid-pachytene stage. Based on these data, our immunostaining of MLH1 and SYCP1 on the meiotic spreads demonstrated that ClpP and ClpX were required for chromosome synapsis. It is known that crossover will not happen if the chromosomes are in a condition of asynapsis[40]. We further showed that the meiotic obstruction eventually led to apoptosis of the cKO spermatocytes, which provided an explanation of the abnormal testicle morphology in the mutant mice. Based on these data, we conclude that ClpP and ClpX are required for the maintenance of spermatocytes, but not the spermatogonia.

The most direct role of mitochondria in meiosis is energy supply, as the various activities of meiosis, such as chromosomal dynamics, require energy to keep these activity ongoing[41]. In meiotic prophase I, the telomeres rather than the centromere assemble the chromosome and bind to the nuclear envelope[42,43]. The previous studies suggest telomere attachment to the nuclear envelope is a prerequisite for subsequent prophase events, including homologous pairing and recombination[44]. Several structural molecules which mediate telomere-nuclear envelope (NE) were identified in recent years, such as Speedy A (cyclin-dependent kinase 2 activator), TERB1 (telomere repeat binding bouquet formation pro-tein 1), SUN1 (Sad1 and UNC84 domain containing 1), KASH5(Klarsicht/ANC-1/Syne/homology 5), TERB2 (telomere repeatbinding bouquet formation protein 2), and MAJIN (membrane-anchored junction protein). Deletion of any one of above proteins in mice displayed impaired telomere-NE attachment[43–48]. Telomere-NE complex assembly is essential for early meiotic prophase I, the impairment of telomere-NE complex assembly results in various of meiotic defects, like accumulated nucleoplasmic telomeres and prophase arrest with aberrant synapsis, these defects could eventually lead to the cell apoptosis[49]. In this study, we demonstrated the deficiency of ClpP or ClpX induced detachment of telomere-NE, the single layer of high-resolution confocal images clearly displayed the dropped telomeres (TRF1 labeled) in the central region of the nuclei. Similar phenomenon was observed in the MFN2 cKO spermatocytes[35]. Although we observed the similar effects with the deletion of telomere-NE structural molecules, however, since *Clpp*, *Clpx* and *Mfn2* are mitochondrial genes rather than meiotic specific genes, we suggested that the impaired telomere-NE attachment might be caused by the damaged functions of mitochondria, such as oxidative phosphorylation and energy supply, which is not in a direct manner. This impaired telomere-NE attachment phenotype might be the real reason why ClpP/ClpX deficient spermatocytes cannot be survived. Therefore, we studied the formation of microtubule in the ClpP/ClpX deleted spermatocytes, as cytoskeleton is known to mediate chromosome dynamics[50]. Surprisingly, we found that ClpP/ClpX deletion disrupted formation of α-tubulin but not β-tubulin. Electron microscopy showed abnormal mitochondrial morphology in the ClpP/ClpX cKO germ cells. The physiological functions of mitochondria were also disrupted in the ClpP/ClpX cKO spermatocytes as demonstrated by two typical criteria of mitochondrial dysfunction, namely, decrease in mitochondrial membrane potential and increase in ROS level. These data clearly demonstrated that the deficiency of ClpP/ClpX impaired mitochondrial functions in germ cells, resulting in disruption of meiosis.

Both the transcriptome analysis and the m6A-sequencing indicate significant alteration of metabolic pathways in the mutant spermatocytes. Besides, the majority of highlighted genes from both high-throughput sequencing analysis are meiosis or germ cell relevant genes, this double confirmed our severe phenotypes in cKO animals for spermatogenesis. These sequencing data were utilized to help us screen out the most altered genes and pathways. After noticing the metabolic pathways might be outstanding in all screened signaling pathways, we then evaluated the activation level for mTOR pathways by accessing the related protein expression. We determined the expression of mTOR signaling and surprisingly found mTORC1 signaling pathway was activated in the ClpP/ClpX deleted spermatocytes. Therefore, we tested whether manipulation of the mTORC1 activity by rapamycin could rescue the effect of ClpP/ClpX deficiency in spermatogenesis and meiosis. Our in vivo rapamycin administration experiment demonstrated the treatment down-regulated mTORC1 signals with decreased phosphorylation of two main substrates of mTORC1, S6K and S6. The increase in the phosphorylation level of 4EBP1 observed after rapamycin injection was consistent with a previous study[32]. These molecular changes contributed to phenotypic changes in the testicular tissue with reappearance of round spermatids and spermatozoa in the rapamycin treated animals. The observations indicate partial rescue of the meiotic process after manipulation of the metabolism in the germ cells. Unfortunately, we could not observe any mature sperm in the epididymis of the treated mice, this might be because signaling pathway(s) other than mTORC1 were altered and affected meiosis in the ClpP/ClpX mutated male germ cells. Thus, rapamycin treatment could not totally rescue all the affected pathways. Indeed, our sequencing data revealed alterations of many other signaling and biological processes upon ClpP/ClpX deficiency. These data support the conclusion that the mTORC1 signaling is over-activated in the ClpP/ClpX deficient spermatocytes.

In summary, our study demonstrated the role of ClpP and ClpX, in spermatogenesis. We showed that they are essential to maintain normal mitochondrial morphology and functions in the

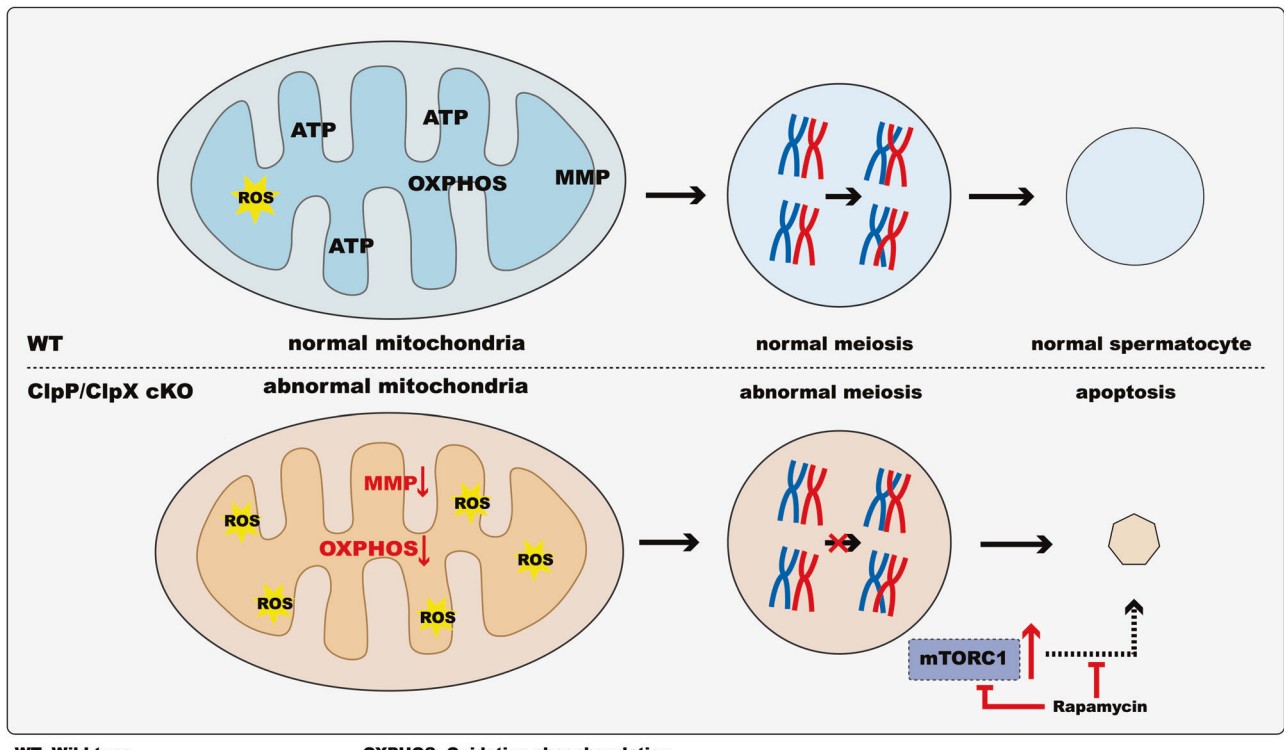

**WT:** Wild type
**cKO:** Conditional knockout
**ATP:** Adenosine triphosphate

**OXPHOS:** Oxidative phosphorylation
**ROS:** Reactive oxygen species
**MMP:** Mitochondrial membrane potential

**Fig. 9 A schematic representation of the proposed model illustrating the role of ClpP/ClpX in spermatogenesis.** ClpP/ClpX deficiency impairs mitochondrial functions and leads to reduced mitochondrial quality in spermatocytes. This deficiency disrupts energy supply during meiosis and attenuates the zygotene-pachytene transformation of spermatocytes. Consequently, dysregulated spermatocytes undergo apoptosis, resulting in decreased testicular size and the formation of vacuolar structures within the seminiferous tubules. Furthermore, the deletion of ClpP/ClpX leads to the over-activation of the mTORC1 pathway in spermatocytes. Long-term inhibition of the mTORC1 signaling, achieved through rapamycin treatment in vivo, partially rescues spermatogenesis. Overall, this model highlights the crucial roles of ClpP and ClpX in regulating mitochondrial functions, meiosis, and spermatogenesis.

male germ cells, and their deletion in the germ cells disrupts meiosis, which eventually leads to apoptosis of spermatocytes and decrease in testicular size. The ClpP or ClpX deficient mice are infertile and lack mature sperms in the testis and epididymis. High-throughput sequencing analysis highlighted alteration of metabolic pathways in the mutant spermatocytes. We confirmed dramatic upregulation of the mTOR signaling in the ClpP/ClpX cKO spermatocytes. Rapamycin in vivo treatment down-regulated the mTORC1 signaling and partially rescued the testicular phenotype of the mutant mice. Our results provide an unique insight of ClpP/ClpX complex in mTOR-mediated metabolism during meiosis and spermatogenesis in mice (Fig. 9).

## Methods

**Animals.** This study have complied with all relevant ethical regulations for animal testing. All handlings and experimental procedures of mice were performed in accordance with the guidelines of the National Health and Medical Research Council of China. All the mice used were of the C57BL/6J background and were kept in the animal facilities at Peking University Shenzhen Hospital. Male 5–8-week C57BL/6J mice were used in the follow experiments. The animals were housed under a controlled environment with free access to water and food, and with lights switched on between 6:00 and 18:00. All experimental protocols were approved by the ethics committee of the University of Hong Kong-Shenzhen Hospital (Permit Number: [2021]035/[2023]035-002). All the biological repeats are not less than 3 if not specifically mentioned.

**Generation of the *Clpp/Clpx* conditional knock-out mouse line and genotyping.** Exon 3-exon 5 of the *Clpp* gene and exon 3 of the *Clpx* gene were targeted for deletion by insertion of two LoxP sites in the mouse genome via the CRISPR/Cas9 and homology directed repair (HDR) techniques (Fig. 1a). Two guide RNAs (gRNA), donor vector containing the LoxP sites, and Cas9 mRNA were co-injected into fertilized mouse eggs to generate the targeted cKO offspring. $F_0$ founder animals were identified by PCR followed by sequencing analysis. The mice were then bred with WT mice to generate the $F_1$ generation and to test for germline transmission. The gRNA sequences used are listed in supplementary Information (Table S3).

The *Clpp/Clpx* floxed mice were identified by PCR analysis with the Rapid Taq polymerase (Cat. P222-02, 2X Rapid Taq Master Mix, Vazyme, Nanjing, China). A small tissue piece was excised at the tail tip of the mice and was lysed with 200 μL of digestion buffer (50 mM KCl, 10 mM pH 9.0 Tris-HCl, 0.1 % Triton X-100 and 0.4 mg/mL Proteinase K) at 55 °C overnight. Then, each sample was incubated at 98 °C for 10 min to denature the Proteinase K and centrifuged at 12,000 rpm for 5 min. The supernatant containing the genomic DNA was then used in the PCR assay. The PCR primers used are listed in supplementary Information (Table S3).

The PCR mixture consisted of 1 μL of mouse tail genomic DNA, 1 μL of forward primer (10 μM), 1 μL of reverse primer (10 μM), 12.5 μL of 2 X Rapid Taq Master Mix, and 9.5 μL of ddH$_2$O. The PCR program involved an initial denaturation step at 95 °C for 3 min followed by 32 cycles comprising denaturation at 95 °C for 15 s, annealing at 60 °C for 15 s, and extension at

72 °C for 15 s. This was followed by a final additional extension step at 72 °C for 5 min. The PCR products were separated by gel electrophoresis using a 1.5% agarose gel. The homozygous ClpP/ClpX floxed mice were then crossed with the *Stra8-Cre* mice, which expressed Cre specifically in the male germ cells[36]. The ClpP/ClpX cKO mice can be identified via the Cre-LoxP system[51]. The sequence of primers for the PCR analysis are listed in supplementary Information (Table S3).

In all experiments, the control mice were chosen from the same littermate of the cKO mice. The control mice were genotyped as *Clpp^{fl/fl}*, *Clpx^{fl/fl}*; homozygous ClpP/ClpX cKO mouse were genotyped as *Clpp^{fl/fl};Stra8-Cre and Clpx^{fl/fl};Stra8-Cre.*

**Tissue collection and histological analysis**. Male mice were euthanized by carbon dioxide inhalation[52]. The testes were immediately collected and fixed in Bouin's solution (Cat. HT10132, Sigma-Aldrich, MO, USA) for 16 h before dehytodration, paraffin embedding, sectioning at 7 μm thick and mounting on glass slides. For histological examination, the tissue sections were de-paraffinized, stained with hematoxylin solution for 90 s, washed three times with ddH_2O, mounted with neutral balsam (Cat. G8590, Solarbio, Beijing, China) and imaged under a light microscope. For indirect immunofluorescence assays, the testes were fixed in 4% paraformaldehyde for 24 h before processing as described above.

**Chromosome spread assays**. The tunica albuginea of the testes was removed and the testicular tissue was transferred into a 1.5 mL RNase/DNase Free tube (Cat. AXYMCT150CS, Axygen, Corning, NY, USA) containing 1 mL PBS (Cat. 10010023, Gibco, Thermo Fisher Scientific, MA, USA). A pair of forceps (Cat. HEC7.1, Tweezers Round, Carl Roth, Karlsruhe, Germany) were used to crush the seminiferous tubules for around 5 min, or until the large tissue pieces had been dissociated. The resulting cell suspension was filtered into a new 1.5 mL microtube though a 40 μm cell strainer (Cat. 352340, Falcon, Corning, NY, USA). The cells were centrifuged at 3500 rpm for 3 min, after which the supernatant was discarded. The pellet was gently resuspended in 1 mL PBS and the cells were washed by centrifugation 1–2 times until the supernatant was clear. After the final wash, the cells were resuspended in 1 mL of hypotonic solution (30 mM Tris HCl, 17 mM trisodium citrate, 5 mM EDTA, 50 mM sucrose, pH 8.8) and incubated at room temperature for 30-40 min. The cells were centrifuged at 3500 rpm for 3 min and the cell pellet was resuspended and incubated in 100 μL sucrose (100 mM) for 5 min at room temperature before fixation by addition of same volume of fixative buffer (0.33 M paraformaldehyde, 100 μL of 10% Triton X-100, 10 mL PBS and 30 μL of 1 N NaOH). Two circles were drawn on glass slides with a PAP pen (Cat. Z377821, Sigma-Aldrich) to mark the location of the specimen, after which 20 μL of the fixed cell suspension was applied to each circle, and the cells were allowed to settle and attach for 3 h at room temperature. Once the cell suspension was completely dry, the slides were stored at −80 °C for up to 3 months prior to the immunostaining.

As to the quantification of meiotic spreads, the slides were imaged under a ZEISS LSM 900 with Airyscan 2 laser scanning confocal microscope, the cells were captured at ×63 magnification. Individual cells were evaluated by the expression of labeled marker proteins and the results were generated and showed in either bar charts or individual dot plots by GraphPad Prism 9. Student's *t*-test were used to show the difference between each two groups. *P*-values less than 0.05 were considered to be statistically significant.

**Immunofluorescence assay**. Fixed cells or tissue sections were permeabilized in phosphate buffered saline (PBS) containing 0.1% Triton X-100 (Cat. T8787, Sigma-Aldrich; PBST) for ~15 min. The cells/tissues were washed twice with PBST, after which they were incubated with blocking PBS buffer containing 5% BSA (Cat. A1933, Sigma-Aldrich, MO, USA) for 1 h at room temperature. The samples were then incubated with the primary antibody in blocking buffer overnight at 4 °C, before washing for 10 min with three changes of PBS and incubation with appropriate Alexa Fluor-tagged secondary antibody (1:500 dilution in blocking buffer) for 1 h at room temperature. The samples were then washed thoroughly with PBST and briefly with distilled H_2O, after which they were mounted with the ProLong Diamond Antifade mountant containing DAPI (Cat. P36962, Invitrogen, MA, USA). The samples were incubated at room temperature overnight to cure the mountant, and then stored at 4 °C prior to examination under a confocal microscope. The antibodies and their dilution used are listed in Supplementary Information (Tables S1 and S2).

Mitochondrial membrane potential was evaluated using the JC-1 probe (Cat. T3168, Invitrogen, MA, USA). In brief, spermatocytes were cultured in DMEM/F12 medium containing 2 mM JC-1 for 30 min at 37 °C and washed for 3 × 3 min with PBS. The samples were immediately imaged by confocal microscopy. JC-1 dye exhibits a potential-dependent accumulation in mitochondria as indicated by an emission shift of fluorescence from green (~529 nm) to red (~590 nm). Thus, mitochondrial depolarization is indicated by a decrease in the red/green fluorescence intensity ratio.

Reactive oxygen species (ROS) in spermatocytes were determined by carboxy-H2DCFDA (Cat. C400, Invitrogen, MA, USA), which is a fluorescent oxidative stress indicator. The spermatocytes were pre-treated with DMEM/F12 medium containing 10 mM H_2O_2 for 5 min. They were then washed and incubated with 10 μM carboxy-H2DCFDA in DMEM/F12 for 30 min at 37 °C, after which they were washed 3 times each for 3 min with PBS and then immediately imaged under a confocal microscope.

**Confocal imaging**. Images of the fluorescence labelled tissue sections or cells were acquired using a ZEISS LSM 900 with Airyscan 2 laser scanning confocal microscope with Hybrid Detectors (HyD). Images were captured with either a Leica HC PL APO CS2 63×/1.4 NA oil immersion objective lens or a Leica HC PL APO 20×/0.7 NA CS2 dry objective lens. Alex Fluor 488, Alex Fluor 546/594, Alex Fluor 647 and DAPI fluorescence were captured with an argon laser operating at 488 nm, a HeNe laser at 561 nm, a HeNe laser at 633 nm, or a diode-pumped solid-state laser at 405 nm, respectively, using 488 nm excitation/519 nm detection, 552 nm excitation/575 nm detection, 633 nm excitation/670 nm detection, and 405 nm excitation/461 nm detection, respectively.

**Purification of male germ cells**. Male mice germ cells were purified following a well-establish protocol[40,53,54]. In brief, mice were sacrificed, and the testes were decapsulated. The seminiferous tubules were isolated from other testicular tissues by incubation with 0.2% W/V collagenase type I (Cat. 17100017, Gibco, MA, USA) and 1 mg/mL DNase I (Cat. A3778, Applichem, IOWA, USA) in a water bath at 37 °C for 10 min, after which the samples were centrifuged at 1200 rpm for 2 min. The pellet containing the seminiferous tubules was resuspended in 5 ml of 0.25% trypsin (Cat.15050057, Gibco, MA, USA) containing 1 mg/mL DNase I at 37 °C for 5 min with gentle shaking. The sample was centrifuged at 1500 rpm for 5 min, after which the pelleted cells were resuspended in 20 mL of high glucose DMEM (Cat. 12100046, Gibco, MA, USA) containing 0.5% BSA

and filtered through a 40 µm nylon cell strainer (Cat. 352340, Falcon®, BD, NJ, USA). The recovered cells in the filtrate were resuspended in 20 mL DMEM containing 0.5% BSA and loaded into a cell separation apparatus (BOMEX Corporate) containing a 2–4% BSA gradient in 600 mL of DMEM. After sedimentation for 3 h, the cells were collected into tubes from the bottom of the separation apparatus at a rate of 10 mL/min. The cell type and purity in each fraction were assessed according to their diameter and morphological characteristics under a light microscope.

**Protein extraction and western blotting analysis.** Purified spermatocytes from WT and ClpP/ClpX cKO male C57BL/6 mice testes were suspended in lysis buffer [50 mM HEPES-KOH (pH 7.5), 100 mM KCl, 2 mM EDTA, 10% glycerol, 0.1% NP-40, 10 mM NaF, 0.25 mM $Na_3VO_4$, and 50 mM β-glycerophosphate] supplemented with complete protease inhibitor (Cat. 04693116001, Roche, Basel, Switzerland). The samples were homogenized and centrifuged at 20,000 g for 20 min at 4 °C, after which the supernatant was retained for western blotting analysis. The proteins in each sample were separated using 4–12% Bis-Tris gels (Cat. M00652, SurePAGE™, GenScript, Nanjing, China) and a mini protein electrophoresis system (Cat. 1658034, BIO-RAD, CA, USA) following the manufacturer's instructions. The protein bands were transferred to polyvinylidene fluoride (PVDF) membranes (Cat. IPVH00010, Immobilon, Millipore, MA, USA) via a Mini Trans-Blot Electrophoretic Transfer Cell (Cat. 1703930, BIO-RAD, CA, USA). The immunoreactive bands were detected and analyzed with a BIO-RAD ChemiDoc MP imaging System (Cat. 12003154, BIO-RAD, CA, USA) in conjunction with the Image Lab Software (BIO-RAD, CA, USA). The semi-quantitively analysis of relative protein levels in each sample were normalized to β-tubulin or β-actin to standardize the loading. The primary and secondary antibodies for immunoblotting are shown in Supplementary Information (Tables S1 and S2).

**Quantitative reverse-transcription polymerase chain reaction (qRT-PCR).** Total RNA was extracted from purified pachytene spermatocytes using BSA gradient sedimentation and was reverse transcribed using the High-capacity cDNA Reverse Transcription kit (Cat. 4368813, Thermo Fisher Scientific, CA, USA) according to the manufacturer's instructions. Five genes (*Sdhb*, *Uqcrc2*, *Atp5a1*, *Ndufv1* and *Cox1*) and the house-keeping gene *Gapdh* were quantified using the real time-qPCR with SYBR® Green master mix (Cat. 1725124, iTaq™ Universal SYBR® Green Supermix, BIO-RAD, CA, USA), and an Applied Biosystems 7500 Real-time PCR system (Cat. 4351107, Applied Biosystems™, Thermo Fisher Scientific, MA, USA). The following program was used: Activation at 95 °C for 3 min (1 cycle) and then 40 cycles of denaturation at 95 °C for 15 s and annealing/extension at 60 °C for 30 s. The data were normalized to GAPDH and the relative levels of mRNA were quantified using the $2^{-\Delta\Delta Ct}$ method. The primers used are listed in supplementary Information (Table S3).

**RNA sequencing and data analysis.** RNA sequencing was performed in an Illumina HiSeq PE150 (Novogene Corp. Inc., Beijing, China). Total RNA was purified from pachytene spermatocytes using the Trizol reagent (Cat. 15596026, TRIzol™, Thermo Fisher Scientific, CA, USA) according to the manufacturer's instructions. The RNA quality and quantity were evaluated using a Bioanalyzer (Agilent Co.) and a Qubit (Life Technologies), respectively, and normalization was performed using the Novomagic software (Novogene Corp. Inc.) to generate the number of reads in fragments per kilobase per million mapped fragments (FPKM). Genes with an average FPKM > 1 in at least one condition were used in subsequent scatterplot analyses. Differential expression analysis

was performed using the DESeq2, and genes with an adjusted *P*-value < 0.05 were considered as being differentially expressed. Expression pattern clusters were generated by unsupervised hierarchical clustering analysis and K-means clustering algorithm using the R. DAVID, and the Ingenuity Pathways Analysis (IPA) were used to reveal the Gene Ontology (GO) and Kyoto Encyclopedia of Genes and Genomes (KEGG) analysis. Three biological replicates were performed for RNA sequencing assays.

**Methylated RNA immunoprecipitation (MeRIP) sequencing and data analysis.** The mRNA of m6A was sequenced by the MeRIP-seq at Novogene (Beijing, China). Briefly, a total of 300 µg RNA was extracted from pachytene spermatocytes. The integrity and concentration of the extracted RNA was determined in an Agilent 2100 bioanalyzer (Agilent) and a simpliNano spectrophotometer (GE Healthcare), respectively. Fragmented mRNA (~100 nt) was incubated for 2 h at 4 °C with anti-m6A polyclonal antibody (Cat. 202-003, Synaptic Systems, Göttingen, Germany) in the immunoprecipitation experiment. The immunoprecipitated mRNAs or Input was used for library construction with the NEB Next ultra-RNA library prepare kit for Illumina (New England Biolabs). The libraries were sequenced on an Illumina Novaseq or Hiseq platform with a paired-end read length of 150 bp according to the standard protocols. The sequencing was carried out with 30 independent biological replicates. Raw data of fastq format were firstly processed using the fastp (version 0.19.11). Reference genome and gene model annotation files were downloaded from the NCBI genome website (https://www.ncbi.nlm.nih.gov). After mapping reads to the reference genome, the exomePeak R package (version 2.16.0) was used for the m6A peak identification in each anti-m6A immunoprecipitation group with the corresponding input samples serving as a control, and a *q*-value threshold of enrichment of 0.05 was used for all data sets. Using the differential peak calling, genes associated with differential peaks were identified followed by GO and KEGG analyses.

**Drug preparation and in vivo pharmacological treatment.** Rapamycin (Cat. HY-10219-25mg, MCE, New Jersey, United States) was firstly dissolved in 1 mL DMSO (Cat. D2650, Sigma, St. Louis, United States) with slight vortexing to prepare a stock concentration of 25 mg/mL. The stock rapamycin solution was stored at −80 °C up to 6 months. Before injection, the stock solution was diluted with PBS to a final injection concentration of 3.125 mg/mL. The ClpP cKO mice and ClpX cKO mice were injected intraperitoneally every other day with 8 mg/kg rapamycin, from PD 10 to PD 35. The control mice from same litter were injected with the same volume of PBS. After rapamycin treatment, the mice were sacrificed and their testicular tissue were collected for molecular and histological evaluation.

**Statistics and reproducibility.** The data were analyzed with the GraphPad Prism 8 using the Student's *t*-test or one-way ANOVA followed by Tukey's test. *P*-values less than 0.05 were considered to be statistically significant. Graphs were generated using Microsoft Excel and figures were prepared with the CorelDraw version X8 (Corel Corp., Ottawa, ON, Canada).

**Reporting summary.** Further information on research design is available in the Nature Portfolio Reporting Summary linked to this article.

## Data availability
All source data for graphs and the uncropped blots in the main text can be found in Supplementary Data and Supplementary Information (Fig. S5, Supplementary Data 1 and Supplementary Data 2). The datasets presented in this study can be found in online

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

## Acknowledgements

Authors would like to express special thanks to Dr. Kui Liu from The University of Hong Kong for his suggestions and guidance in germ cell isolation and meiosis studies. Thanks also go to Dr. Guangxin Li from Peking University Shenzhen Hospital for his kind discussion on the work of mTOR signaling pathways. This work was supported by Shenzhen Science and Technology Program, China (No. RCYX20200714114705073); Shenzhen Fundamental Research Program, China (No. JCYJ2020010915042414); Shenzhen Science and Technology Program, China (No. KQTD20190929172749226); National Natural Science Foundation of China (No. 81971453).

## Author contributions

Conceptualization: C.G., T.W. Methodology: C.G., T.W. Investigation: C.G., X.Y., J.G., J. Zheng, R.H. Animal breeding: C.G., X.Y., J.G., Z. Hu, R.H., Z. Hai, J.S. Funding acquisition: W.Y., J. Zhang, T.W. Project administration: C.G., T.W. Supervision: C.G., T.W. Writing – original draft: C.G. Writing – review & editing: C.G., W.Y., P.Z., J.G. T.W.

## Competing interests

The authors declare no competing interests.
