## [Peer Review File · Communications Biology]

Reviewers' comments:

Reviewer #1 (Remarks to the Author):

This work studies the functions of two mitochondria protein, ClpP and ClpX in the male meiotic prophase I. The authors propose that ClpP/ClpX is required for maintaining mitochondrial functions in spermatocytes during meiosis and spermatogenesis. The study is generally well designed and the conclusions are generally supported by the experimental evidence. This study is absolutely needed for the field in understanding how exactly mitochondria are involved in meiosis I in mammalian spermatocytes. However, there are several issues that need to be addressed before the paper can be considered for publication.

Major point: the work needs a mechanistic explanation in terms of how mitochondria genes take part in meiosis. Although lots of phenotype description is presented, the author should draw a conclusion or a hypothesis from their data how ClpP and ClpX affect meiosis mechanistically. More specifically, which substage of prophase I do these genes affect? Do mitochondria affect DSB repair, or crossover formation, or synapsis?

Minor points:

1. Fig. 1., PCR genotyping is not needed for the paper. WB is a semi-quantitative method, so its quantification is not needed, or indicate that it is semi-quantitative.
2. Fig. 2., the emptiness but not the diameters matter for KO mice. So I suggest to delete the diameter analysis.
3. Fig. 3. The abnormal SC may be better presented by some insets with higher magnification. Fig. Fv and Fvi are redundant as the apoptosis of spermatocytes are obvious, and the quantified timepoints may represent a stage where many cells have disappeared.
4. Fig. 4. Can the authors discuss why some spermatocytes still process to diplotene-like, if ClpP and ClpX are so important? This will help the readers to understand the functions of mitochondria genes in prophase I.
5. Fig. 4. What is the significance of lacking MLH1 when ClpP and ClpX are knocked out? Please discuss.
6. Fig. 5Aiv, the TRF1 signals seem clustered, is this a coincidence, or it has some significance?
7. The telomere-NE detachment should be better discussed. Is this the true reason why the cells die?
8. Fig. 7 presented the sequencing results and has interesting implications. The authors needs to validate by RT-PCR for some of the genes selected.
9. It is very interesting to link mTOR activation to the KO phenotypes. Can the authors explain why elevated mTOR that should enhance metabolism and growth accompany cell death? This may open up a new understanding of how signalling is involved in meiosis.

Reviewer #2 (Remarks to the Author):

This manuscript is entitled "ClpP/ClpX deficiency impairs mitochondrial functions and mTORC1 signaling during spermatogenesis and meiosis." This title, along with the abstract, describe the main

points of the manuscript very well. However, this reviewer recommends that the words "and meiosis" from the title. Meiosis is a part of spermatogenesis and this manuscript does not address female meiosis and thus, it is not a general conclusion that meiosis is altered.

Overall, the manuscript is succinctly, well-written with a particularly good introduction. The experiments address key points and concepts of spermatogenesis through a series of solid, classical methods with addition of some innovation especially in the use of rapamycin treatment to test rescue through inhibition of mTORC1.

A few points to address:

1) There is a need for some editing due to unusual or unclear wording. A few examples follow but there are quite a number of wording irregularities throughout:

A. Line 147: "determined" should be "examined."

B. Line 154: Use of the term "meiotic spreads assay" should be "meiotic spread assays."

C. Line 342: Remove "The."

D. Line 361: Rewrite for clarity.

E. Line 377: This sentence is not accurate perhaps as the link to ATP has not been shown in this manuscript and mitochondria perform numerous functions.

2) The discussion is mostly a recapitulation of results and does not add much to the manuscript. The authors should include a model figure and discuss the meaning of the results more. For example, why does rapamycin result in partial rescue; what are the authors ideas on further studies, clinical applications, and other extensions of the study.

3) The transcriptome profiling and m6A-seq analysis is superficial. It may be useful but great discussion beyond GO analysis and shared-gene expression diagrams.

4) The quantification of the meiotic spreads (methods) should be included and include the statistical methods.

Below, we provide a point-by-point response to the reviewers' comments and highlight the modifications made in the revised manuscript:

Reviewers' comments:

Reviewer #1 (Remarks to the Author):

This work studies the functions of two mitochondria protein, ClpP and ClpX in the male meiotic prophase I. The authors propose that ClpP/ClpX is required for maintaining mitochondrial functions in spermatocytes during meiosis and spermatogenesis. The study is generally well designed and the conclusions are generally supported by the experimental evidence. This study is absolutely needed for the field in understanding how exactly mitochondria are involved in meiosis I in mammalian spermatocytes. However, there are several issues that need to be addressed before the paper can be considered for publication.

Major point: the work needs a mechanistic explanation in terms of how mitochondria genes take part in meiosis. Although lots of phenotype description is presented, the author should draw a conclusion or a hypothesis from their data how ClpP and ClpX affect meiosis mechanistically. More specifically, which substage of prophase I do these genes affect? Do mitochondria affect DSB repair, or crossover formation, or synapsis?

-Response: Thanks for your comments and helpful suggestions. To sum up, we believe the deletion of ClpX or ClpP caused severe damage of mitochondrial functions with affected energy production, which further affect prophase I activities starting from abnormal DSB repair and asynapsis. Moreover, the incomplete synapsis consequently results to the disappearance of cross-over activity among chromosomes. We have also added a graph figure to summarize our results in revised manuscript according to Reviewer 2' s comments. Please see Fig. 9.

Minor points:

1. Fig. 1., PCR genotyping is not needed for the paper. WB is a semi-quantitative method, so its quantification is not needed, or indicate that it is semi-quantitative.

-Response: Thanks for these suggestions. We have deleted the PCR genotyping image and we have stated in the methods and results that WB is semi-quantitative statistical analysis in the revised manuscript.

2. Fig. 2., the emptiness but not the diameters matter for KO mice. So I suggest to delete the diameter analysis.

-Response: Thanks for your comments and nice suggestions, we deleted the diameter analysis in the figure 2 in revised manuscript.

3. Fig. 3. The abnormal SC may be better presented by some insets with higher magnification. Fig. Fv and Fvi are redundant as the apoptosis of spermatocytes are obvious, and the quantified timepoints may represent a stage where many cells have disappeared.

-Response: Thanks for your suggestions, we accept your comments and provide a higher resolution zoomed-in image in the Fig3. We have deleted the Fig3. Cv-Cvi and Fv-Fvi in revised manuscript as well.

4. Fig. 4. Can the authors discuss why some spermatocytes still process to diplotene-like, if ClpP and ClpX are so important? This will help the readers to understand the functions of mitochondria genes in prophase I.

-Response: Thanks for the question. We now add several sentences in revised manuscript from Line 416 to discuss the relationship between the diplotene-like spermatocytes and ClpP/X deletion. "Interestingly, we noticed that there are diplotene-like meiotic cells, shared a limited proportion, in *Clpx/Clpp*-deficient spermatocytes. The sister chromosomes in these diplotene-like cells are gradually separated in both ends of chromosomes, while the strong γ H2A.X retention in autosomes illustrate these cells are diplotene-like rather than normal diplotene cells. We also interested why we can detect diplotene-like cells in ClpX/ClpP cKO spermatocytes, the reason of it might not be alone. We suggest that although the function of mitochondria got disrupted in ClpX/ClpP cKO spermatocytes, the cell themselves must have their own mechanism to compensate the mitochondrial functions or energy supply to favor the continuation of meiotic process. This could be one possible reason that why we can still detect a little proportion of cells in diplotene-like stage from ClpX/P cKO spermatocytes. Therefore, it also suggests that *Clpx* and *Clpp* might not directly regulate the meiotic process, instead, the deletion of these genes directly damage mitochondria, and affect meiosis in a indirectly way." This might help to understand the specific role of these two mitochondrial genes in prophase I of meiosis.

5. Fig. 4. What is the significance of lacking MLH1 when ClpP and ClpX are knocked out? Please discuss.

-Response: Thanks for point it out, we now add more discussion from Line 434 "MLH1, which is started to be expressed on chromosomes after crossover activity, it is also a marker protein of the crossover occurring. The lack of expression of MLH1 in both ClpX cKO and ClpP cKO spermatocytes means there is little crossover occurred in the mid-pachytene stage." on the significance of lacking MLH1 expression in ClpX/P cKO cell.

6. Fig. 5Aiv, the TRF1 signals seem clustered, is this a coincidence, or it has some significance?

-Response: Thanks for this question. We believe it is a coincidence as in the low magnification representative and other pictures, the majority of TRF1 signals were not in a cluster manner in ClpP cKO spermatocytes (Figure 5Aiii).

7. The telomere-NE detachment should be better discussed. Is this the true reason why the cells die?

-Response: Thanks for your comments and nice suggestions, we add a paragraph from Line 448 to 460 to discuss this part of issue in revised manuscript.

8. Fig. 7 presented the sequencing results and has interesting implications. The authors needs to validate by RT-PCR for some of the genes selected.

-Response: We would like to express our gratitude for your valuable comments. We carefully considered your suggestion to validate the selected genes using RT-PCR. However, we would like to provide a rationale for not performing RT-PCR validation in our study. Our research focuses on investigating the broader landscape of gene expression patterns within a specific pathway, rather than focusing on individual gene expression levels. By utilizing RNA-sequencing technology, we were able to obtain a comprehensive overview of the gene expression profile and identify key pathways that are potentially involved in the biological process under investigation. And we further performed western blot analysis to validate the expression at protein level instead of on transcript-level validation.

Performing RT-PCR validation for a subset of genes may provide additional confirmation of their expression levels, but we are thinking it may not significantly contribute to the main objective of our study. We believe that the comprehensive RNA-sequencing data we have presented is sufficient to support our findings and conclusions regarding the pathway analysis. Nonetheless, we appreciate your suggestion and will discuss the possibility of conducting RT-PCR validation for individual gene expression patterns in future research, where a gene-centric analysis is more appropriate.

If you still strongly recommend conducting the RT-PCR validation, we kindly request further clarification regarding the specific genes that you suggest for validation. Understanding your perspective on which genes you believe would be crucial for validation will help us better address your concerns and evaluate the feasibility of conducting the suggested experiments.

9. It is very interesting to link mTOR activation to the KO phenotypes. Can the authors explain why elevated mTOR that should enhance metabolism and growth accompany cell death? This may open up a new understanding of how signalling is involved in meiosis.

-Response: We appreciate your insightful comments. While mTOR is generally known for its role in promoting metabolism and growth, its precise function and effects can vary depending on the cellular context and specific conditions. One possible explanation for the association between elevated mTOR activation and cell death in our study is the dysregulation of energy homeostasis. Spermatocytes undergoing meiosis require a delicate balance of energy supply to sustain the demanding processes of chromosomal recombination and segregation. The impaired mitochondrial functions resulting from ClpP/ClpX deficiency may disrupt energy production and supply, leading to an energy crisis within the spermatocytes. Under such conditions, the over-activation of mTORC1 signaling, which typically stimulates metabolic

pathways, might exacerbate the energy imbalance, pushing the cells toward apoptosis rather than promoting their survival and growth. However, the exact mechanisms underlying the association between elevated mTOR activation and cell death in the context of ClpP/ClpX deficiency during spermatogenesis remain to be fully elucidated. This intriguing observation highlights the complexity of signaling networks and their involvement in meiotic processes. Further investigations are warranted to unravel the precise molecular mechanisms that link ClpP/ClpX deficiency, mTORC1 signaling, and cell fate decisions in spermatocytes.

Reviewer #2 (Remarks to the Author):

This manuscript is entitled “ClpP/ClpX deficiency impairs mitochondrial functions and mTORC1 signaling during spermatogenesis and meiosis.” This title, along with the abstract, describe the main points of the manuscript very well. However, this reviewer recommends that the words “and meiosis” from the title. Meiosis is a part of spermatogenesis and this manuscript does not address female meiosis and thus, it is not a general conclusion that meiosis is altered.

-Response: That's true and we are agree with you that the meiosis should be deleted in the title. Thanks a lot for this suggestion.

Overall, the manuscript is succinctly, well-written with a particularly good introduction. The experiments address key points and concepts of spermatogenesis through a series of solid, classical methods with addition of some innovation especially in the use of rapamycin treatment to test rescue through inhibition of mTORC1.

A few points to address:

- 1) There is a need for some editing due to unusual or unclear wording. A few examples follow but there are quite a number of wording irregularities throughout:
 - A. Line 147: “determined” should be “examined.”
 - B. Line 154: Use of the term “meiotic spreads assay” should be “meiotic spread assays.”
 - C. Line 342: Remove “The.”
 - D. Line 361: Rewrite for clarity.
 - E. Line 377: This sentence is not accurate perhaps as the link to ATP has not been shown in this manuscript and mitochondria perform numerous functions.

-Response: Thanks for pointing out these issues, we have already modified it accordingly in the revised manuscript.

- 2) The discussion is mostly a recapitulation of results and does not add much to the manuscript. The authors should include a model figure and discuss the meaning of the results more.

-Response: We appreciate your feedback and suggestions on our manuscript. We acknowledge your concern and have addressed it by significantly revising the discussion section to provide a more in-depth analysis and interpretation of the results. We have expanded our discussion on the implications and significance of the findings, considering the broader context of meiosis and spermatogenesis. We have also emphasized the novel insights gained from our study, particularly regarding the roles of ClpP and ClpX in regulating meiosis and spermatogenesis. We have also added a new model figure as Fig. 9 to conclude our results in revised manuscript.

3) The transcriptome profiling and m6A-seq analysis is superficial. It may be useful but great discussion beyond GO analysis and shared-gene expression diagrams.

-Response: Thanks for your comments. In this study, transcriptome profiling and m6A-seq analysis were employed to identify altered signaling pathways. While we acknowledge that the initial presentation of these results may have appeared superficial, we want to emphasize that these analyses were crucial in identifying key genes and pathways involved in our study. Notably, the KEGG and GO analyses highlighted numerous genes, with a significant proportion belonging to the spermatogenesis gene family. This observation strengthens the relevance of our phenotypic findings in spermatogenesis.

To delve deeper into the mechanisms and implications of our findings, we have now expanded our discussion beyond the initial GO analysis and shared-gene expression diagrams. We have provided a more in-depth analysis of the identified genes and pathways, emphasizing their functional significance in spermatogenesis and their potential contributions to the observed phenotypes. We have incorporated additional literature references and highlighted the biological relevance and implications of the altered signaling pathways.

Furthermore, based on the findings from the sequencing data, mTOR signaling emerged as a key pathway of interest in both the RNA-seq and m6A-seq analyses. To further evaluate the significance of mTOR signaling, we examined the protein levels within the mTOR pathway. This additional analysis strengthens the link between our transcriptome profiling, m6A-seq results, and the functional relevance of mTOR signaling in spermatogenesis. We have now included a detailed discussion of these findings and their implications in the revised manuscript.

4) The quantification of the meiotic spreads (methods) should be included and include the statistical methods.

-Response: Thanks for your suggestions, we added a short paragraph to mention the quantification and statistical method for meiotic spreads.

REVIEWERS' COMMENTS:

Reviewer #1 (Remarks to the Author):

The author has already addressed all my question.